# The genomic legacy of aurochs hybridisation in ancient and modern Iberian cattle

Torsten Günther[1]*, Jacob Chisausky[1], Ángeles M Galindo-Pellicena[2], Eneko Iriarte[3], Oscar Cortes Gardyn[4], Paulina G Eusebi[4], Rebeca García-González[3], Irene Ureña[5], Marta Moreno-García[6], Alfonso Alday[7], Manuel Rojo[8], Amalia Pérez[3], Cristina Tejedor Rodríguez[8], Iñigo García Martínez de Lagrán[9], Juan Luis Arsuaga[4], José-Miguel Carretero[3,10], Anders Götherström[5], Colin Smith[3,11]*, Cristina Valdiosera[3,11,12]*

[1]Human Evolution, Department of Organismal Biology, Uppsala University, Uppsala, Sweden; [2]Centro Mixto UCM-ISCIII de Evolución y Comportamiento Humanos, Madrid, Spain; [3]Laboratorio de Evolución Humana, Universidad de Burgos, Burgos, Spain; [4]Universidad Complutense de Madrid, Madrid, Spain; [5]Centre for Palaeogenetics, Stockholm, Sweden; [6]Instituto de Historia – CSIC, Madrid, Spain; [7]Área de Prehistoria, University of the Basque Country, Bilbao, Spain; [8]Department of Prehistory and Archaeology, Valladolid University, Valladolid, Spain; [9]Departamento de Prehistoria y Arqueología, UNED, Madrid, Spain; [10]Unidad Asociada de I+D+i al CSIC Vidrio y Materiales del Patrimonio Cultural (VIMPAC), Burgos, Australia; [11]Department of Archaeology and History, La Trobe University, Melbourne, Australia; [12]CENIEH (Centro Nacional de Investigación sobre la Evolución Humana), Burgos, Spain

*For correspondence:
torsten.gunther@ebc.uu.se (TG);
cismith@ubu.es (CS);
cvaldioser@gmail.com (CV)

Competing interest: The authors declare that no competing interests exist.

## eLife Assessment

Using genomic data from ancient and modern samples, this **important** study investigates the genomic history of cattle in Iberia, focusing on the admixture between domestic cattle and their wild ancestors, aurochs. The authors present **convincing** evidence for interbreeding between domestic cattle and wild aurochs since the Neolithic period, although the evidence of sex-biased introgression is weak. The authors also show that the aurochs ancestry in cattle stabilized at ~20% since ~4000 years ago and continues into modern breeds; however, the aurochs ancestry is not heightened in a modern breed of Spanish fighting bulls that are bred for aggressiveness. The work will be of interest to evolutionary biologists and quantitative geneticists who seek to understand the genomic history and genetic basis of trait variation of domesticated animals.

**Abstract** Cattle (*Bos taurus*) play an important role in the life of humans in the Iberian Peninsula not just as a food source but also in cultural events. When domestic cattle were first introduced to Iberia, wild aurochs (*Bos primigenius*) were still present, leaving ample opportunity for mating (whether intended by farmers or not). Using a temporal bioarchaeological dataset covering eight millennia, we trace gene flow between the two groups. Our results show frequent hybridisation during the Neolithic and Chalcolithic, likely reflecting a mix of hunting and herding or relatively unmanaged herds, with mostly male aurochs and female domestic cattle involved. This is supported by isotopic evidence consistent with ecological niche sharing, with only a few domestic cattle possibly being managed. The proportion of aurochs ancestry in domestic cattle remains relatively

constant from about 4000 years ago, probably due to herd management and selection against first generation hybrids, coinciding with other cultural transitions. The constant level of wild ancestry (~20%) continues into modern Western European breeds including Iberian cattle selected for aggressiveness and fighting ability. This study illuminates the genomic impact of human actions and wild introgression in the establishment of cattle as one of the most important domestic species today.

## Introduction

Domestication of livestock and crops has been the dominant and most enduring innovation of the transition from a hunter-gathering lifestyle to farming societies. It represents the direct exploitation of genetic diversity of wild plants and animals for human benefit. Ancient DNA (aDNA) has proved crucial to understanding the domestication process and the interaction between domesticated species and their wild relatives both within domestication centres and throughout the regions that the domestics expanded into (*Frantz et al., 2019*; *Frantz et al., 2020*; *Botigué et al., 2017*; *Irving-Pease et al., 2018*; *Verdugo et al., 2019*; *Daly et al., 2018*; *Daly et al., 2021*; *Fages et al., 2019*; *Librado et al., 2021*; *Yurtman et al., 2021*; *Bergström et al., 2020*; *Ginja et al., 2023*; *Larsson et al., 2024*; *Kaptan et al., 2024*). The origins of the European domestic taurine, *Bos taurus,* are located in the Fertile Crescent (*Peters et al., 1999*; *Helmer et al., 2005*) and unlike dogs, pigs, and goats, where the wild forms are still extant, the wild cow (the aurochs) went extinct in 1627. Aurochs, *Bos primigenius*, was present throughout much of Eurasia and Africa before the expansion of domestic cattle from the Levant that accompanied the first farmers during the Neolithisation of Europe. Upon arrival, these early incoming domesticates inevitably coexisted with their wild counterparts in great parts of Europe facilitating gene flow in both directions. In general, taxa within the genus *Bos* can hybridise and produce fertile offspring (*Zhang et al., 2020*) which may have facilitated and contributed to domestication, local adaptation, and even speciation (*Verdugo et al., 2019*; *Palacio et al., 2017*; *Wecek et al., 2017*; *Ward et al., 2022*). Mitochondrial DNA studies have previously indicated gene flow between domestic cattle and aurochs outside their domestication centre (*Beja-Pereira et al., 2006*; *Achilli et al., 2008*; *Schibler et al., 2014*; *Cubric-Curik et al., 2022*; *Bro-Jørgensen et al., 2018*) and more recently, genomic studies have shown the presence of European aurochs ancestry in modern taurine cattle breeds (*Park et al., 2015*; *Upadhyay et al., 2017*; *Rossi et al., 2024*). Although cattle have represented a significant economic resource and a prominent cultural icon for millennia, and have been studied through aDNA for more than a decade (*Verdugo et al., 2019*; *Beja-Pereira et al., 2006*; *Achilli et al., 2008*; *Park et al., 2015*; *Rossi et al., 2024*; *Anderung et al., 2005*; *Erven et al., 2024*), our understanding of the interaction of early cattle herds and wild aurochs is still limited due to a lack of time-series genomic data. This gap of knowledge includes European aurochs' genetic contribution to modern domestic breeds and human management of these animals in the past.

Aurochs have been widely exploited by humans since the European Palaeolithic and archaeological evidence indicates that the species survived in Europe until historical times. Iberia could have served as a glacial refugium for aurochs (*Rossi et al., 2024*), and the most recent evidence for aurochs is found at a Roman site in the Basque Country (*Altuna, 2002*). Domestic cattle were introduced into Iberia with the Mediterranean Neolithic expansion and reached the northern coast of the peninsula around 7000 years cal BP (*Cubas et al., 2016*). Consequently, aurochs and domestic cattle have coexisted in Iberia for about five millennia. Since then, cattle have played an important role in Iberian societies as a source of food and labour, as well as cultural events such as bullfighting. Currently, there are more than 50 bovine breeds officially recognised in the Iberian Peninsula including the Lidia breed, a primitive, isolated population selected for centuries to develop agonistic-aggressive responses with the exclusive purpose of taking part in such socio-cultural events (*Cañón et al., 2008*). Recently, it has been reported that Lidia breed individuals have the largest brain size among a comprehensive dataset of European domestic cattle breeds and are the most similar to wild aurochs (*Balcarcel et al., 2021*). The combination of aggressiveness and larger brain size in the Lidia breed may suggest a higher proportion of aurochs ancestry compared to other cattle breeds.

Here, we present the genomes and stable isotope data of Iberian Bovine specimens ranging from the Mesolithic into Roman times from four archaeological sites (*Figure 1A*). We explore the extent of interbreeding between wild aurochs and domestic cattle over time and the correlation of genetic

**eLife digest** For over five thousand years, domesticated cows and oxen in the Iberian Peninsula lived alongside their wild counterparts, the aurochs. These large and aggressive animals, from which modern European cattle descends, only went extinct during the 17th century. Genetic evidence points to aurochs and livestock having interbred during their long coexistence; when and how these mixing events took place, however, remains unclear. Details regarding the management of ancient herds are also missing.

To address these questions, Günther et al. analysed the DNA extracted from ancient bovine bones sampled at four Iberic archaeological sites. This revealed that wild aurochs and cattle frequently interbred during the last 8,000 years. Mating principally took place between male aurochs and domesticated cows but slowed down after 4,000 years, resulting in modern cattle having inherited about 20% of genes from their wild relatives. This percentage was consistent across various breeds, including one renowned for its aggressivity and which has been selected for centuries for Spanish bullfighting.

Additional bone analyses revealed that aurochs and ancient cattle shared comparable diets composed primarily of wild vegetation. Only some domestic animals showed signs of having been fed crops.

These findings help us understand how modern cattle breeds came to be. The genes they inherited from aurochs may help them survive harsh environmental conditions, such as extreme heat or diseases. In the future, researchers could use this knowledge to refine breeding programs.

ancestry with metric identification and ecological niches. Finally, we compare the results to genomic data obtained from modern Iberian cattle breeds to estimate the genetic contribution of the now-extinct aurochs to the Iberian farming economy.

## Results
### Exploratory analysis
We successfully sequenced 24 bovine specimens excavated at four prehistoric sites in Iberia (*Figure 1A*). Nine of these individuals were inferred to or suspected to represent aurochs based on morphology or chronology. Direct radiocarbon dates and contextual dating placed the individuals between the Mesolithic (oldest sample moo001, 8641–8450 cal BP) and the Roman Age (youngest sample moo010p, 2260–2150 cal BP). It should be noted that while all post-Mesolithic samples were found at archaeological sites with evidence for herding of other domestic fauna such as ovicaprids (Appendix 1), we do not know whether these bovids were herded or hunted. Based on the number of reads mapping to the X chromosome, 13 individuals were identified as female and 11 as male, for the samples with sufficient amounts of reads for this analysis. Sequencing coverage of the nuclear cattle genome was low to medium, reaching up to 4.7× with a mean of 0.38× (*Supplementary file 1*). The sequence data for non-UDG-treated libraries showed damage patterns characteristic of aDNA (*Appendix 2—figures 1–3*). Based on reads mapping to the mitochondrial genome, we were able to estimate contamination for 11 samples with most samples showing low levels (<2%) of contamination. One individual (moo013a) showed a high contamination of 51.7% [33.5, 69.9] and was excluded from further analysis. One Mesolithic individual (moo040) showed 8.3% [3.1, 13.5] contamination, which was included in the initial exploratory analysis but not used for the analysis of hybridisation between wild and domestic as this analysis only focused on post-Mesolithic individuals.

Nine individuals were assigned to the mitochondrial P1 haplogroup, one to haplogroup T1, one to haplogroup T4, and 12 to haplogroup T3 (*Supplementary file 1*). P haplogroups are dominant among and thought to be endemic to European aurochs (*Sinding and Gilbert, 2016*), but are occurring at low frequencies in modern European cattle breeds (*Cubric-Curik et al., 2022*). The prevalence of the T3 haplogroup in our samples is expected; this haplogroup is dominant among modern European *B. taurus* and is the most common haplogroup in ancient Western European domestic cattle. T3 was found in directly dated Neolithic samples from different sites providing direct evidence for the arrival of domestic cattle in northern Iberia during the Neolithic (*Supplementary file 1*). The specimen assigned to T1 (moo009a) is notable since this individual was previously used to argue for Bronze Age

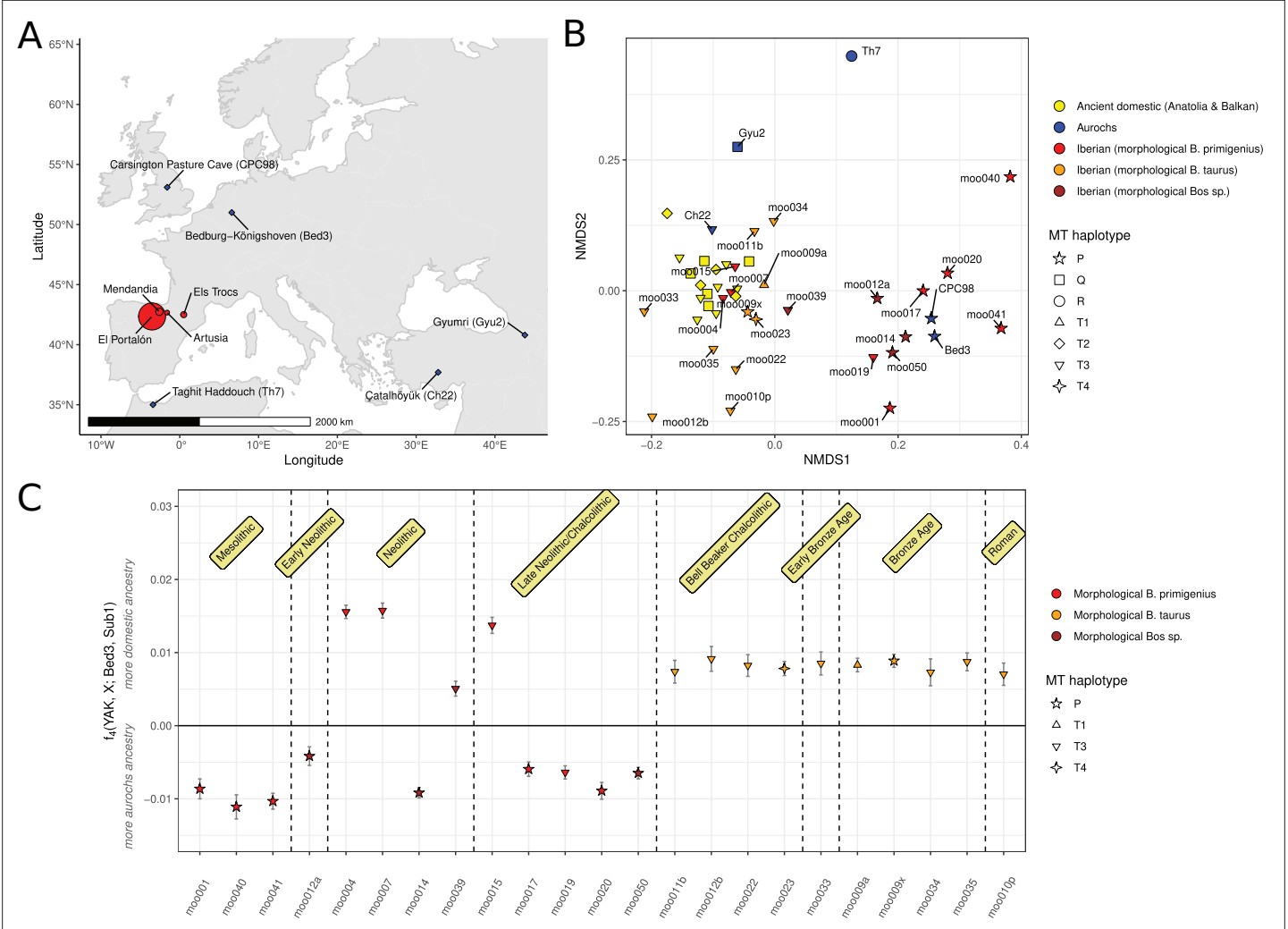

**Figure 1.** Data overview and descriptive ancestry analysis. (**A**) Map of Europe showing the Iberian sampling sites (red circles, size proportional to sample sizes) and the sites for published aurochs genomes used in the analysis (blue diamonds). (**B**) Non-metric multi-dimensional scaling (NMDS) ordination of nuclear data of Iberian samples considered *B. taurus* samples (orange), Iberian samples that were morphologically considered *B. primigenius* (red), other Iberian *Bos* samples (brown), ancient domestic cattle from the Balkans and Anatolia (yellow), and aurochs (blue). Data point shape corresponds to mitochondrial haplogroups. (**C**) $f_4$ statistic measuring allele sharing of the Iberian samples with European aurochs (Bed3) or Anatolian Neolithic cattle (Sub1). Error bars indicate 95% confidence intervals estimated using a block jackknife procedure. The time periods displayed are contextual.

contact between Iberia and Africa, where the T1 haplogroup is thought to have originated (*Anderung et al., 2005*; *Colominas et al., 2015*). T4 is usually considered to be restricted to Asian breeds with rare finds in Europe, restricted to the Balkans (*Cubric-Curik et al., 2022*). The presence of T4 in Chalcolithic Iberia suggests that this haplogroup must have been distributed across Western Europe at low frequencies in prehistory. Furthermore, the fact that some specimens that were morphologically identified as aurochs carry domestic T haplogroups implies some level of interbreeding between the two groups.

As mitochondrial genomes only reflect the maternal line of ancestry, they are not informative about the exact extent of interbreeding in our dataset. To avoid being constrained by the variation in modern domestic breeds as with common approaches such as projected principal component analysis (PCA) (*Appendix 2—figure 4*), we performed non-metric multi-dimensional scaling (NMDS) ordinations on a matrix of pairwise outgroup $f_3$ statistics to explore the genomic ancestry of the sequenced individuals. For reference, we included early cattle genomes from Anatolia and the Balkans as well as aurochs excavated from Morocco (Th7), Armenia (Gyu2), Anatolia (Ch22), Germany (Bed3), and Britain (CPC98) (*Verdugo et al., 2019*; *Park et al., 2015*; *Erven et al., 2024*), and calculated pairwise

outgroup $f_3$ statistics. The NMDS ordination outcome (*Figure 1B*) seems to represent a separation between domestic autosomal ancestry (to the left) and European aurochs ancestry (to the right). In contrast, aurochs from other regions (Th7 and Gyu2) seem genetically distinct. Many early domestic samples from Iberia fall close to early cattle from the Balkans and Anatolia as well as the Anatolian aurochs (Ch22). Notably, at least two of the Iberian samples in this cluster (moo004, moo007) were morphologically identified as aurochs. Eight of the nine Iberian samples with haplogroup P fall to the right in the plot, together with the aurochs from Germany (Bed3) and Britain (CPC98). Additionally, one individual carrying a domestic T3 mitochondrial genome (moo019) appears closer to the aurochs samples than the domestics. Out of nine samples that were presumed aurochs based on their morphological features, only six would be considered aurochs based on this analysis. This highlights a substantial overlap between measurements or criteria that are used to distinguish wild and domestic *Bos* based on morphometrics.

This analysis suggests that one can use other European aurochs such as the German Bed3 (Bedburg-Königshoven, 11802–11326 cal BP) (*Erven et al., 2024*) or the British CPC98 (Carsington Pasture Cave, 6874–6602 cal BP) (*Park et al., 2015*) as a reference for Western European aurochs as they seem similar to our three low coverage Mesolithic Iberian samples. This is also supported by a recent parallel study concluding that all Western European aurochs form a clade, possibly even originating from an Iberian glacial refugium (*Rossi et al., 2024*). Using Sub1 (Suberde Höyük, 8171–7974 cal BP) (*Verdugo et al., 2019*), a Neolithic domestic Anatolian individual, and the higher coverage aurochs Bed3 as references, we can perform $f_4$ statistics to measure which Iberian individuals share more alleles with one or the other (*Figure 1C*). Despite the relatively low coverage of some samples, the $f_4$ statistics are highly correlated with the first axis of the NMDS ($R^2$=0.84, p=8.3×10$^{-10}$) implying that they detect the same pattern. Non-overlapping confidence intervals also confirm that the high genetic differentiation between Western European aurochs and domestic cattle allows confident assignment even with low coverage data. The three Mesolithic individuals as well as an additional six, up until the Late Neolithic/Chalcolithic, share most of their alleles with aurochs. Three individuals from the Neolithic and Late Neolithic/Chalcolithic share most of their alleles with domestic Anatolian cattle while two individuals (moo012a and moo039) are more intermediate, suggesting that there could have been some level of hybridisation. More recent samples from the Bell Beaker period onwards all appear to have similar amounts of allele sharing with mostly domestic ancestry but some level of aurochs introgression.

## Quantifying the extent of introgression

While $f_4$ statistics measure allele sharing it does not directly quantify the amount of introgression in the different specimens, hence, we employed three different frameworks to estimate ancestry proportions: $f_4$ ratio (*Patterson et al., 2012*), qpAdm (*Haak et al., 2015*; *Harney et al., 2021*), and Struct-f4 (*Librado and Orlando, 2022*) to model each Iberian individual from European aurochs (Bed3) and/or Anatolian Neolithic cattle (Sub1) as sources (*Table 1*). While the $f_4$ ratio provides a straightforward-to-interpret estimate of aurochs ancestry under a simple two-source model, we also include qpAdm due to the potential of rejecting models and hinting at additional ancestries. We also include Struct-f4 for better samples (>0.1×) as it is more flexible than qpAdm not requiring a strict separation between sources and outgroup populations. While quantitative estimates of European aurochs ancestry for the 20 post-Mesolithic individuals are somewhat correlated between $f_4$ ratio and qpAdm (Spearman's correlation coefficient rho = 0.57, p=0.01), they differ for certain individuals. This highlights differences between the methods, their assumptions about the relationships of sources and outgroups, and their sensitivity to low coverage data. For most parts of this study, we decide to present the $f_4$ ratio results but it is important to highlight that our interpretations are based on the general pattern and not on the ancestry estimates for single individuals.

Most of the 20 post-Mesolithic individuals show indications of both domestic and European aurochs ancestries (*Table 1*). Only three individuals ($f_4$ ratio) or one individual (qpAdm) do not show significant proportions of aurochs ancestry while only one individual ($f_4$ ratio) or three individuals show not significant proportions of domestic ancestry. Furthermore, qpAdm and Struct-f4 suggest low proportions of additional, eastern ancestries represented either by indicine cattle or the Caucasus aurochs Gyu2 in these analyses. While these ancestries are not well resolved and usually have high standard errors, they suggest that multiple western Asian populations contributed to the European early domestic

**Table 1.** European aurochs ancestry proportions in post-Mesolithic Iberian *Bos* samples.

Square brackets are showing block-jackknife estimates of the 95% confidence interval. $f_4$ ratio and qpAdm are using Bed3 as source of European aurochs ancestry unless noted otherwise. Footnotes are added when deviations from the two-source model were needed. Struct-f4 was run in semi-supervised mode to estimate ancestry in the Iberian samples with K=5 as the different ancestries separated at this point. Only individuals with at least 0.1× coverage were included in this analysis to ensure convergence. LNCA = Late Neolithic/Chalcolithic.

| Sample ID | Site | Period | Date cal BP | $f_4$ ratio | qpAdm | Struct-f4 (K=5) |
|---|---|---|---|---|---|---|
| moo012a | El Portalón | Early Neolithic | *contextual* | 0.908 [0.743, 1.072] | 0.928 [0.77, 1.085] | – |
| moo004 | Els Trocs | Neolithic | 7152–6890 | 0.103 [0.025, 0.182] | 0.06 [0.026, 0.098]* | 0.01[¶] |
| moo007 | Els Trocs | Neolithic | 7151–6890 | −0.089 [-0.204, 0.026] | 0.073 [0.033, 0.113]* | 0.039[¶] |
| moo014 | El Portalón | Neolithic | 6491–6403 | 0.861 [0.791, 0.931] | 0.942 [0.905, 0.979][†] | 0.942[¶] |
| moo039 | Mendandia | Neolithic | 7426–7280 | 0.435 [0.334, 0.536] | 0.866 [0.76, 0.971] | – |
| moo015 | El Portalón | LNCA | 5041–4842 | −0.055 [-0.182, 0.072] | 0.444 [0.32, 0.565] | – |
| moo017 | El Portalón | LNCA | 5567–5326 | 0.756 [0.646, 0.866] | 0.811 [0.769, 0.853]* | – |
| moo019 | El Portalón | LNCA | 5556–5325 | 0.723 [0.628, 0.818] | 0.828 [0.786, 0.87]* | 0.654 |
| moo020 | El Portalón | LNCA | *contextual* | 0.775 [0.652, 0.897] | 0.944 [0.80, 1.085] | – |
| moo050 | El Portalón | LNCA | 5468–5320 | 0.788 [0.709, 0.866] | 0.809 [0.773, 0.845]* | 0.663** |
| moo011b | El Portalón | Bell Beaker | *contextual* | 0.109 [-0.097, 0.316] | 0.667 [0.47, 0.865] | – |
| moo012b | El Portalón | Bell Beaker | 4421–4291 | 0.706 [0.528, 0.883] | 0.326 [0.258, 0.394]* | – |
| moo022 | El Portalón | Bell Beaker | *contextual* | 0.169 [0.006, 0.332] | 0.729 [0.55, 0.910] | – |
| moo023 | El Portalón | Bell Beaker | 4153–3976 | 0.179 [0.073, 0.285] | 0.641 [0.54, 0.738] | 0.21[¶] |
| moo033 | El Portalón | Early Bronze Age | *contextual* | 0.241 [0.050, 0.432] | 0.987 [0.79, 1.182] | – |
| moo009a | El Portalón | Bronze Age | 3884–3635 | 0.399 [0.297, 0.502] | 0.284 [0.244, 0.324]* | 0.152[¶] |
| moo009x | El Portalón | Bronze Age | 3829–3513 | 0.198 [0.115, 0.280] | 0.259 [0.203, 0.315][‡] | 0.176[¶] |
| moo034 | El Portalón | Bronze Age | 3811–3492 | 0.547 [0.300, 0.794] | 0.112 [-0.061, 0.285][§] | – |
| moo035 | El Portalón | Bronze Age | *contextual* | 0.307 [0.170, 0.445] | 0.821 [0.68, 0.963] | – |
| moo010p | El Portalón | Roman | 2334–2156 | 0.397 [0.205, 0.589] | 0.281 [0.217, 0.345]* | – |

*To produce a fitting and feasible model (p>0.01) a minor contribution (≤ 5%) of indicine ancestry is required to fit the data.

[†]Model does not fit with Bed3 as European aurochs source but fits well (p=0.49) when using CPC98.

[‡]To produce a fitting and feasible model (p>0.01) a contribution of 33.7% Caucasus aurochs ancestry (Gyu2) is required to fit the data. This additional source is not well resolved as the standard error is large (21.2%).

[§]To produce a fitting and feasible model (p>0.01) a contribution of 85.9% Caucasus aurochs ancestry (Gyu2) is required to fit the data. This additional source is not well resolved as the standard error is large (74.6%).

[¶]Struct-f4 also assigns a small proportion of Caucasus aurochs ancestry (Gyu2) to this individual (<5%).

**Struct-f4 also assigns a proportion of Caucasus aurochs ancestry (Gyu2) to this individual (14.0%).

gene pool. Notably, most Neolithic and pre-Bell Beaker Chalcolithic individuals show either predominantly domestic or aurochs ancestry while many Bell Beaker and Bronze Age individuals show more intermediate values of aurochs ancestry. In fact, from the Bronze Age onwards, most estimates overlap with the approximately 25% aurochs ancestry in modern Iberian cattle (*Figure 2*; *Appendix 2—table 1*; *Eusebi et al., 2022*).

A limitation of this analysis is the availability of genomes that can be used as representatives of the source populations. We used German and British aurochs to represent Western European aurochs ancestry and a single Anatolian Neolithic to represent the original domestic cattle that was introduced into Europe. Our Mesolithic Iberian aurochs contained too little endogenous DNA to be used as a proxy aurochs reference and all Neolithic and Chalcolithic samples estimated with predominantly aurochs ancestry (including the 2.7× genome of moo014) already carry low (but significant) levels of domestic ancestry. However, the fact that all of these aurochs samples carried P mitochondria strongly

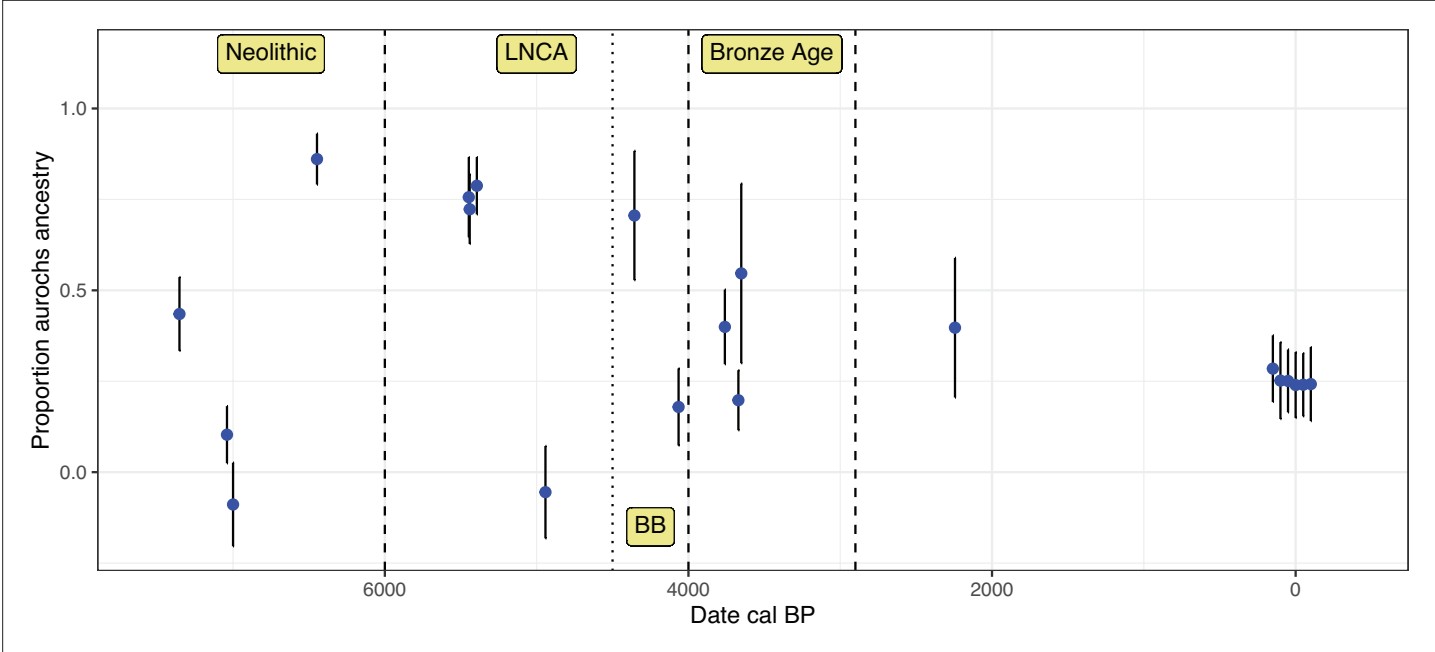

**Figure 2.** Development of aurochs ancestry over time. Estimates of aurochs ancestry (estimated using the $f_4$ ratio with Bed3 as European aurochs source) in directly dated post-Mesolithic Iberian samples over time. Error bars indicate the 95% confidence interval estimated using a block-jackknife procedure. Modern Iberian whole-genome sequenced Lidia individuals are added around date 0 with some horizontal jitter. Approximate boundaries for the main sampling periods are indicated by dashed vertical lines.

suggests that Western European aurochs can be considered monophyletic. Furthermore, a recent parallel study also concluded that all Western European aurochs form a clade (*Rossi et al., 2024*). The Anatolian Sub1 might also not be depleted of any European aurochs ancestry and could not fully represent the original European Neolithic gene pool as also indicated by qpAdm and Struct-f4 identifying small proportions of other Asian ancestries in some Iberian individuals. While these caveats should affect our quantitative estimates of European aurochs ancestry, they should not drive the qualitative pattern as our tests would still detect any excess European aurochs ancestry that was not present in Neolithic Anatolia.

An important question that remains unexamined is the exact process that led to the hybridisation since this could provide insight into human management practices or, more generally speaking, mating patterns between wild and domestic individuals. The fact that some individuals with predominantly aurochs ancestry carry T haplogroups (moo019) and that some individuals with predominantly domestic ancestry carry P haplogroups (moo009x) implies that females contributed in both directions. To assess whether the admixture process was sex-biased, we compared aurochs ancestry patterns on the X chromosome and autosomes (*Figure 3*). Since females carry two X chromosomes and males only have one, we can assume that an excess of a certain ancestry on the X chromosome indicates more females from that particular source population. While the estimates are noisy due to the low coverage data and even less sites available for the X chromosome, it is striking that all but one individual with mostly domestic autosomal ancestry (>50%) show even lower point estimates of aurochs ancestry on the X chromosome. This pattern even extends into the modern Iberian individuals. Male-biased aurochs introgression has been suggested based on mitochondrial haplotypes before (*Verdugo et al., 2019*). In the absence of aurochs Y chromosomal data, however, it is difficult to assess sex-biased processes from uniparental data alone. The comparison of X chromosomes and autosomes should theoretically have more power to detect such processes as they are less sensitive to genetic drift due to their recombining nature (*Keinan et al., 2009*) but estimation of ancestry proportions on the X chromosome can be affected by different biases (*Lazaridis and Reich, 2017*; *Goldberg et al., 2017*; *Pfennig and Lachance, 2023*). Overall, our results are consistent with previous observations that the contribution of wild ancestry into domestic cattle was mostly through aurochs bulls.

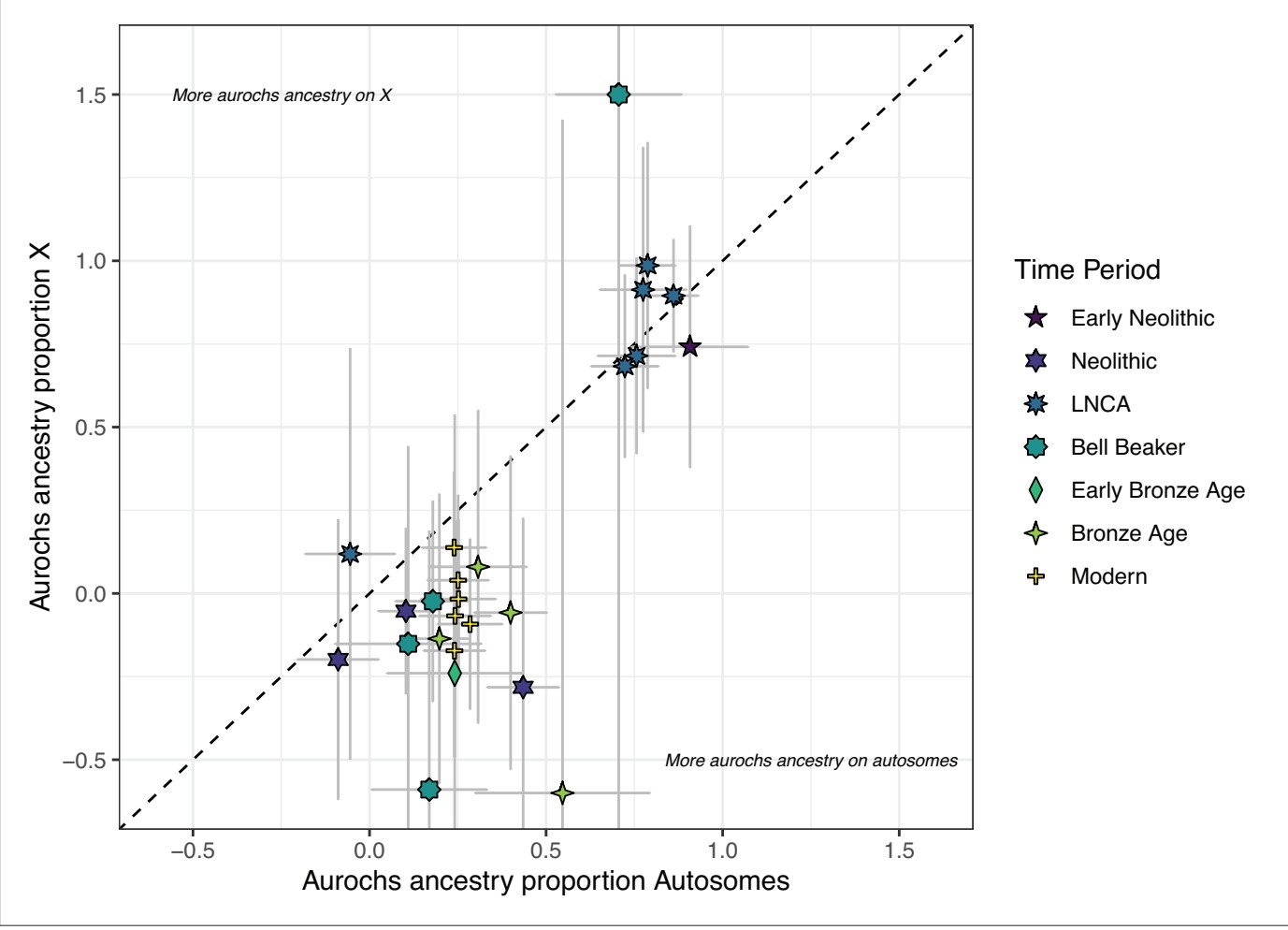

**Figure 3.** Comparison of f₄ ratio estimated aurochs ancestry of post-Mesolithic Iberian samples on the autosomes vs X chromosomes. Error bars indicate the 95% confidence interval estimated using a block jackknife procedure.

## Aurochs ancestry in modern breeds and the Spanish Lidia cattle breed

We estimated aurochs ancestry in a set of Western European cattle breeds (*Upadhyay et al., 2017*) as we performed for the prehistoric samples. Previous studies have used *D* statistics for pairwise comparisons between breeds (*Park et al., 2015*; *Upadhyay et al., 2017*; *da Fonseca et al., 2019*). Such *D* statistics, however, are sensitive to biases including gene flow from populations not included in the analysis (*Rogers and Bohlender, 2015*). Furthermore, qpAdm provides the possibility to reject scenarios not fitting the data. Our point estimates for the aurochs ancestry range between 20% and 30% across all breeds (*Figure 4*) and do not show an increase in aurochs ancestry in Iberian breeds (*da Fonseca et al., 2019*). This result differs from the previous studies which suggested geographic differences in Western and Central Europe and we believe this could be due to ancestry from other, non-European groups in some commercial breeds (Appendix 2). Importantly, not all tested breeds did fit the simple two-source model Anatolian Neolithic domestic+European aurochs, likely representing low levels of contributions from other groups, e.g., indicine cattle (*Upadhyay et al., 2017*). The presence of indicine ancestry can be confirmed in a qpAdm analysis using three sources resulting in fitting models for all breeds (*Appendix 2—table 4*).

Cattle have played an important role in Iberian culture during the last centuries as they have been part of numerous traditional popular events including bullfighting. The Lidia breed, a heterogeneous group of Iberian cattle that is mainly bred for aggressive behaviour, has commonly been used for such popular festivities (*Cañón et al., 2008*). Even though Lidia cattle has only been actively bred for agonistic behaviour for about 200 years, some people attribute their aggressiveness and appearance

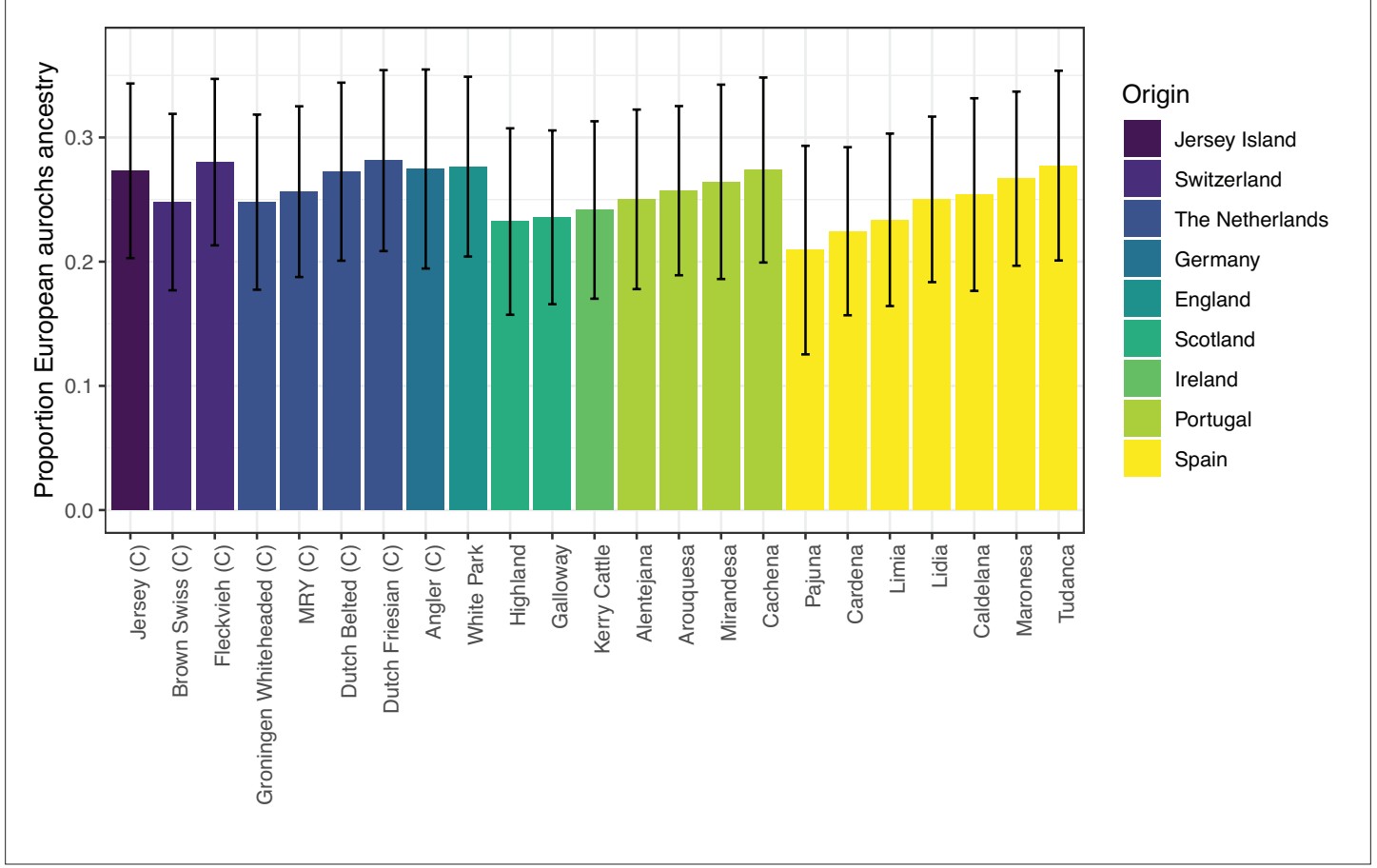

**Figure 4.** qpAdm estimates of aurochs ancestry in modern Western European cattle breeds from the *Upadhyay et al., 2017*, dataset. Commercial breeds are marked with a 'C'. The figure is only showing breeds with feasible and non-rejected two-source models, as all results are shown in *Appendix 2—table 1*. Error bars are showing block-jackknife estimates of the 95% confidence interval.

The online version of this article includes the following figure supplement(s) for figure 4:

**Figure supplement 1.** qpAdm aurochs ancestry estimates for the six individual Lidia genomes.

as an indication of high levels of aurochs ancestry (*Balcarcel et al., 2021*). We additionally use medium coverage genomes of six Lidia individuals (*Eusebi et al., 2022*) to estimate their proportion of aurochs ancestry. Lidia cattle in the *Upadhyay et al., 2017*, dataset had a point estimate of 25% (95% confidence interval: [18.5, 31.5]) aurochs ancestry and estimates in the individual genomes ranged from 17.6% [10.9, 24.3] to 23.5% [17.0, 30.0] (*Figure 4—figure supplement 1*) – all overlapping with the observed range for other Western European breeds. Despite some variation between individuals, which might be attributable to noise due to low coverage sequencing data in the reference populations, we do not observe a systematically elevated level of aurochs ancestry compared to other modern breeds or ancient samples since the Bronze Age. While these results reject the idea that the specifics of Lidia cattle can be attributed to a substantially increased genome-wide aurochs ancestry, it does not rule out the possibility that the roots of their aggressiveness and appearance are indeed due to aurochs variants at key loci responsible for those traits. An in-depth investigation of such questions would require a larger dataset of aurochs genomes as well as a more comprehensive Lidia sampling due to their fragmentation in highly distant genetic lineages (*Cañón et al., 2008*).

## Stable isotope analysis

In addition to their ancestry, we studied the ecology of the bovids through stable isotope analysis of bone collagen. *Lynch et al., 2008*, suggested that stable isotope data could be used to infer niche separation between the species in Britain, with domestic cattle in more open settings, while aurochs (about 1‰ more depleted in $\delta^{13}$C) were habitually in more forested areas, or wet ground. This is most

likely facilitated through human management of the domestic cattle, separating them from their wild counterparts. In contrast, *Noe-Nygaard et al., 2005*, failed to observe such an effect in samples from Denmark and northern Germany.

Considering our dataset and other data published on Iberian cattle (categorised on morphology/date) (*Fernández-Crespo and Schulting, 2017*; *Fontanals-Coll et al., 2015*; *Garcia Guixé et al., 2006*; *Jones et al., 2019*; *Navarrete et al., 2017*; *Salazar-García, 2009*; *Salazar-García, 2011*; *Salazar-García et al., 2014*; *Sarasketa-Gartzia et al., 2018*; *Villalba-Mouco et al., 2018*), we observe that the nitrogen isotope means are statistically different only when our data are compared using morphological characteristics, not genetic distinctions (see *Supplementary file 1* and Appendix 3). This difference is mostly due to some domestic cattle with δ15N values greater than 6.5‰. This could be explained by some taurine cattle having exclusive habitual access to high nitrogen isotope ratio resources. For example, human management such as corralling on manured ground, or feeding with manured crops, would produce this effect. Nevertheless, there is generally a large amount of overlap in the isotope values for the two groups suggesting that wild and domesticated groups often did not occupy different niches in Iberia.

## Discussion

We generated and analysed biomolecular data from *B. primigenius* and *B. taurus* spanning more than 9000 years in the same region. Cattle are important livestock in the Iberian Peninsula today, and our results illustrate the interaction between domestic cattle and their wild relatives in the past. The two groups show signs of frequent hybridisation starting soon after the arrival of cattle to the peninsula, as evident in our oldest directly dated Neolithic individual (moo039, 7426–7280 cal BP) where signals of carrying both ancestries are clear. Throughout the Neolithic, we observed large variations in the wild vs domestic ancestry per individual, but this pattern later stabilised (to 20–30% aurochs ancestry) from the Chalcolithic/Bronze Age onwards. As we do not know whether the sequenced individuals were hunted or herded, this could reflect a transition from hunting and herding to predominantly herding and it is possible that systematic herd management led to the nearly constant levels of aurochs ancestry over the last 4000 years. This period also coincides with several other societal changes; including the Bell Beaker complex and the introduction of human ancestry from the Pontic steppe into the Iberian Peninsula (*Olalde et al., 2018*; *Valdiosera et al., 2018*; *Olalde et al., 2019*). Around this time, humans also started processing a significantly higher amount of dairy products connected with the 'secondary product revolution' (*Francés-Negro et al., 2021*; *Evershed et al., 2022*). Aurochs were probably present in Iberia until Roman times (*Altuna, 2002*) leaving possibilities for interbreeding but we cannot exclude that various factors such as hunting or changing vegetation had led to a substantial decline in the wild aurochs population around the early Bronze Age. A previous study on cattle morphology from the site of El Portalón described a decrease in size from the Neolithic to the Chalcolithic and a further significant size decrease from the Chalcolithic to the Bronze Age (*Galindo-Pellicena et al., 2020*) and associated this change in size to the aridification of the area at this time (*Martínez-Pillado et al., 2014*). Indeed, this climatic change could also be related to a reduction of the aurochs population contributing to the stabilisation of the levels of ancestry in domestic cattle from the Bronze Age to the present. Nonetheless, our stable isotope results suggest that wild and domesticated groups often did not occupy substantially different niches on the Iberian Peninsula. Material excavated from Denmark suggested that aurochs changed their niches over time (*Noe-Nygaard et al., 2005*) demonstrating some flexibility depending on local vegetation and the possibility of aurochs adapting to changing environments.

The reduced level of aurochs ancestry on the X chromosome (compared to the autosomes) in admixed individuals suggests that it was mostly aurochs males who contributed wild ancestry to domestic herds, a process that had been suggested based on the distribution of mitochondrial haplotypes before (*Verdugo et al., 2019*). A recent parallel study based using ancient genomes also detected male-biased aurochs introgression using similar methods as our study (*Rossi et al., 2024*). Consequently, the offspring of wild bulls and domestic cows could be born into and integrated within managed herds. It is unclear how much of this process was intentional but the possibility of a wild bull inseminating a domestic cow without becoming part of the herd suggests that some level of incidental interbreeding was possible. For Neolithic Turkey, it has been suggested that allowing insemination of domesticated females by wild bulls was intentional, maybe even ritual (*Peters et al., 2012*). Modern

breeders are still mostly exchanging bulls or sperm to improve their stock which manifests in a lower between-breed differentiation on the X chromosome (*da Fonseca et al., 2019*).

The lack of correlation between genomic, stable isotope, and morphological data highlights the difficulties of identifying and defining aurochs to the exclusion of domestic cattle. All of these data measure different aspects of an individual: their ancestry, ecology, or appearance, respectively. While they can give some indication, none of them are a direct measurement of how these cattle were recognised by prehistoric humans or whether they were herded or hunted. It remains unclear whether our ancestry inferences had any correlation to how prehistoric herds were managed and how much intentional breeding is behind the observed pattern of hybridisation. It is even possible that all hybrids identified in this study were part of domestic herds.

Even though wild aurochs populations went extinct, European aurochs ancestry survived into modern cattle with a relatively uniform distribution across Western European breeds. Isolated Iberian Lidia, bred for their aggressiveness, appears to be no exception to this pattern. This rejects the notion that an overall increased proportion of aurochs ancestry causes the distinctiveness of certain breeds, but considering the functional relevance of archaic introgression into modern humans (*Reilly et al., 2022*), it is possible that aurochs variants at functional loci may have a substantial influence on the characteristics of modern cattle breeds. Our low coverage sequencing data did not allow us to investigate this but future bioarchaeological studies combining different types of data will have the possibility to clarify the role of the extinct aurochs ancestry in modern domestic cattle.

## Conclusions

Using a bioarchaeological approach we have demonstrated that since cattle arrived in Iberia there has been hybridisation with the local aurochs population, and that mainly aurochs bulls contributed to the gene pool still found in domestic herds today. Admixture proportions vary for the first few millennia but stabilise during the Bronze Age at approximately 20–30% of wild ancestry in the individuals found at the Iberian archaeological sites, a level that is still observed in modern Iberian breeds, including the more aggressive Lidia breed. This development could be the result of an initial mix of hunting and herding together with a generally loose management of herds, becoming more controlled over time in combination with a reduced importance of hunting wild aurochs.

The amount of hybridisation observed in the ancient cattle makes it difficult to genetically define what a domestic or wild *Bos* is, bringing into doubt the validity of such categorisations. Our interpretation is made more difficult by the overlap in morphological and metric data, creating further difficulties in species determination (especially in hybrids) and niche sharing as revealed by stable isotopes. To some extent, our interpretation is moot, as the salient matter is, how did prehistoric humans interact with cattle? What was their sense of wild and domestic and hybridisation? While we have recognised individual hybrids, to what extent these were part of domestic herds or intentionally bred and managed is uncertain.

Another source of uncertainty in our determinations is the limited knowledge about the genetic diversity in European aurochs. Further regional (and temporally longitudinal) aurochs genomes would aid future genomic studies defining the genetic variation in the European aurochs population.

# Materials and methods
## Data generation

We attempted DNA extractions of 50 archaeological remains from which we successfully extracted DNA from 24 individuals identified as domestic cattle and aurochs excavated from four prehistoric sites in Iberia: El Portalón de Cueva Mayor (n=18), Artusia (n=1), Els Trocs (n=2), and Mendandia (n=3). Teeth and bones were UV irradiated (6 J/cm$^2$ at 254 nm) and the first millimeter of bone/tooth surface abraded using a Dremel tool. DNA was extracted in a dedicated aDNA facility using a silica-based DNA extraction protocol (*Dabney et al., 2013*). For each sample, 100–200 mg of bone or tooth powder were incubated for 24 hr at 37°C, using the MinElute column Zymo extender assembly replaced by the High Pure Extender Assembly (Roche High Pure Viral Nucleic Acid Large Vol. Kit) and performed twice for each sample. DNA extracts were subjected to UDG treatment for the removal of deaminated cytosines and were further converted into blunt-end double-stranded Illumina multiplex sequencing libraries (*Meyer and Kircher, 2010*). Between seven and fifteen qPCR cycles were

performed to amplify the DNA libraries using indexed primers (*Meyer and Kircher, 2010*). These were subsequently pooled at equimolar concentrations and shotgun sequenced on Illumina HiSeq and NovaSeq sequencing platforms.

## Radiocarbon dates

Eight bone and teeth were directly radiocarbon dated (AMS) at Waikato University in New Zealand and two teeth at Beta Analytics in the United States. Radiocarbon dates were calibrated using the OXcal 4.4 program (*Bronk Ramsey, 2009*) and the IntCal20 calibration curve (*Reimer et al., 2020*). Three samples from the site of Mendandia were conventionally radiocarbon dated at Groningen (Netherlands) radiocarbon laboratory and calibrated as above.

## Stable isotopes analysis

Many of the samples analysed here were radiocarbon dated and stable isotope data (via IRMS) were generated in this process, to augment this data we also produced stable isotope data for some additional samples in this dataset, where they were available. The additional samples underwent bone collagen or tooth dentine collagen extraction at the Laboratorio de Evolución Humana (Universidad de Burgos) following the protocol of *Richards and Hedges, 1999*. In brief, this is a cold acid demineralisation, followed by Milli-Q water rinsing, gelatinisation at pH 3 (24 hr at 70°C), Ezee filtering, and lyophilisation. Collagen yields (as % mass of starting bone) were recorded. Stable isotope values ($\delta^{13}$C, $\delta^{15}$N) and %C, %N were measured in duplicate at the Universitat Autònoma de Barcelona, unless only one sample was successful in the analysis. Collagen samples (approx. 0.4 mg) were analysed using a Flash IRMS elemental analyser coupled to a Delta V Advantage isotope ratio mass spectrometer (IRMS), both from Thermo Scientific (Bremen, Germany) at the Institute of Environmental Science and Technology of the Universitat Autònoma de Barcelona (ICTA-UAB). International laboratory standard IAEA-600 was used, with measurements made relative to Vienna PeeDee Belemnite (V-PDB) for $\delta^{13}$C, and air $N_2$ (AIR) for $\delta^{15}$N. The average analytical error was <0.2‰ (1σ) as determined from the duplicate analyses of $\delta^{13}$C and $\delta^{15}$N. In-house standards used was dog hair collected and homogenised for interlaboratory comparisons.

## Data processing

HiSeq X10 reads have been trimmed and merged using AdapterRemoval (*Schubert et al., 2016*) while adapters for NovaSeq 6000 reads have been trimmed with cutadapt (*Martin, 2011*) and merging was performed with FLASH (*Magoč and Salzberg, 2011*) requiring a minimum overlap of 11 bp. Single-end reads of at least 35 bp length were then mapped to the cattle reference genomes UMD3.1 (*Zimin et al., 2009*) and Btau5 (*Cattle Genome Sequencing International Consortium, 2015*) using bwa (*Li and Durbin, 2009*) with the non-default parameters: -l 16500, -n 0.01, and -o 2. Different sequencing runs per sample were merged with samtools (*Li et al., 2009*) and consensus sequences were called for duplicate sequences with identical start and end coordinates (*Kircher and Clifton, 2012*). Finally, reads with more than 10% mismatches to the reference genome were removed. Biological sex was assigned to the samples mapped to the Btau_5 reference genome (as UMD3.1 does not contain a Y chromosome assembly) using the $R_x$ method (*Mittnik et al., 2016*) modified for 29 autosomes.

Mitochondrial contamination was estimated following the approach used by *Green et al., 2008*, for hominins. We first identified nearly private mutations (less than 5% frequency in the 278 diverse mitogenomes used by MitoToolPy and dometree *Peng et al., 2015*, obtained from Dryad, https://doi.org/10.5061/dryad.cc5kn) in each individual and then used the proportion of non-consensus alleles at these sites to estimate contamination. We restricted this analysis to sites with at least 10× coverage, a minimum base quality of 30. Furthermore, transition sites with a C or G in the consensus mitogenome were excluded to avoid over-estimation due to post-mortem damage. Standard errors were estimated assuming a binomial distribution around the point estimate. Code used for this step can be found at https://github.com/GuntherLab/mt_contam_domestic_green, copy archived at *Günther, 2024*.

For comparative purposes, we also processed published data from *Verdugo et al., 2019*; *Palacio et al., 2017*; *Wecek et al., 2017*; *Park et al., 2015*, using the same bioinformatic pipeline. Furthermore, we downloaded sequence data for six Spanish Lidia cattle (*Eusebi et al., 2022*), a single modern water buffalo (*Bubalus bubalis*, Jaffrabadi-0845) (*Dutta et al., 2020*) and a single zebu cattle individual (Sha_3b) (*Chen et al., 2018*) and processed them with our aDNA mapping pipeline. To obtain

a pseudohaploid Yak (*Bos grunniens*) sequence, we followed the approach by *Upadhyay et al., 2017*, splitting the Yak reference genome (*Hu et al., 2012*) contigs into 100 bp fragments and mapping them to the UMD3.1 reference genome.

## Data analysis

Mitochondrial consensus sequences were called using ANGSD (*Korneliussen et al., 2014*) and the options -doFasta 2 -doCounts 1 -minQ 30 -minMapQ 30. Mitochondrial haplogroups were then assigned to the whole mitogenome sequences using the Python script MitoToolPy (*Peng et al., 2015*).

For population genomic analysis, we used a panel of SNPs derived from Run6 of the 1000 genomes project (*Daetwyler et al., 2014*; *Hayes and Daetwyler, 2019*). We obtained a list of SNPs from *Naval-Sánchez et al., 2020*, and reduced the panel to biallelic SNPs of at least 10% minor allele frequency in the joint European *B. taurus*/Asian *Bos indicus* dataset. Prior to genotype calling, all ancient BAM files were modified such that Ts in the first 5 bases of each fragment and As at the last 5 base pairs of each fragment have a base quality of 2. This approach allows to include more sites than excluding all transitions which are potentially affected by post-mortem damage. Prior to genotype calling, all ancient BAM files were modified such that Ts in the first 5 bases of each fragment and As at the last 5 base pairs of each fragment have a base quality of 2. As this is an approach that is not widely used, we compared its effect on downstream analysis. As our main analyses are all based on f statistics, we compared $f_4$ statistics (*Figure 5*) and $f_4$ ratios (*Figure 6*) of our rescaled base quality data with data only using transversion sites. While estimates are highly correlated, the dataset reduced to transversions produces larger confidence intervals in $f_4$ ratios due to the lower number of sites (*Figure 6*). Consequently, we decided to use the rescaled data for all analyses displayed in the main figures.

To generate pseudohaploid representations of each individual, we randomly draw a single read with mapping and base quality of at least 30 at each SNP position. If the allele carried by the ancient individual was not one of the two known alleles, we removed the site from the panel. Using this approach, ~9.1 million autosomal and 248K X chromosomal SNPs were genotyped in the ancient samples. To compare the ancient samples to a diverse set of modern cattle, we used the panel of modern European breeds presented by *Upadhyay et al., 2017*, which were genotyped at ~770,000 SNPs. The ancient samples were genotyped the same way as for the 1000 Bulls project SNP panel.

To conduct an ordination of the nuclear data, sequences of 43 ancient Eurasian cattle and two aurochs were obtained from *Park et al., 2015*, and *Verdugo et al., 2019*. Outgroup $f_3$ statistics were calculated for all pairs of our Iberian *Bos* samples, using a Yak (*B. grunniens*) genome as an outgroup, and a distance matrix for all samples was calculated as 1 $f_3$. All f statistics were calculated in R version 4.1.2 (*R Development Core Team, 2016*) package 'admixtools2' (*Maier et al., 2023*). The distance matrix was used to compute scores for NMDS ordinations using the metaMDS function in the 'vegan' R package and 10,000 random starts (*Oksanen et al., 2020*).

European aurochs introgression α into Iberian individual X was estimated using $f_4$ ratios calculated with POPSTATS (*Skoglund et al., 2015*) and the equation

$$\alpha = 1 - \frac{f_4\left(Gyu2, YAK; X, EuropeanAurochs\right)}{f_4\left(Gyu2, YAK; Sub1, EuropeanAurochs\right)}$$

Both Bed3 and CPC98 were separately tested as aurochs source and Bed3 was chosen for the results presented in the article due to lower confidence intervals. POPSTATS was run with the non-default options –ratio, –testpop, and –not23 to allow for more autosomes than humans have. We also used admixtools2 (*Maier et al., 2023*) and qpAdm (*Haak et al., 2015*; *Harney et al., 2021*) to model the ancestry proportions in the samples. Bed3 was used as a source for European aurochs ancestry (due to lower standard errors in the $f_4$ ratios) while the domestic Anatolian Neolithic Sub1 was used as a source for domesticated cattle ancestry. As 'right' populations, we used Gyu2, *B. indicus*, Yak and *Bison bonasus bonasus* PLANTA. qpAdm was run with auto_only = FALSE, maxmiss = 0.5 and allsnps = TRUE. When the two-source model did not fit (p<0.01) or produced infeasible admixture proportions outside [0, 1], we used rotate_models and qpadm_multi to find alternative models adding CPC98 as an additional possible source or 'right' population. qpAdm was also used for the modern Western European breed panel from *Upadhyay et al., 2017*, adding Bes2 (*Verdugo et al., 2019*) to the 'right' populations and excluding breeds from Italy and the Balkan from the targets as non-taurine ancestry (*Upadhyay et al., 2017*) in them would lead to a rejection of the models. Finally, we

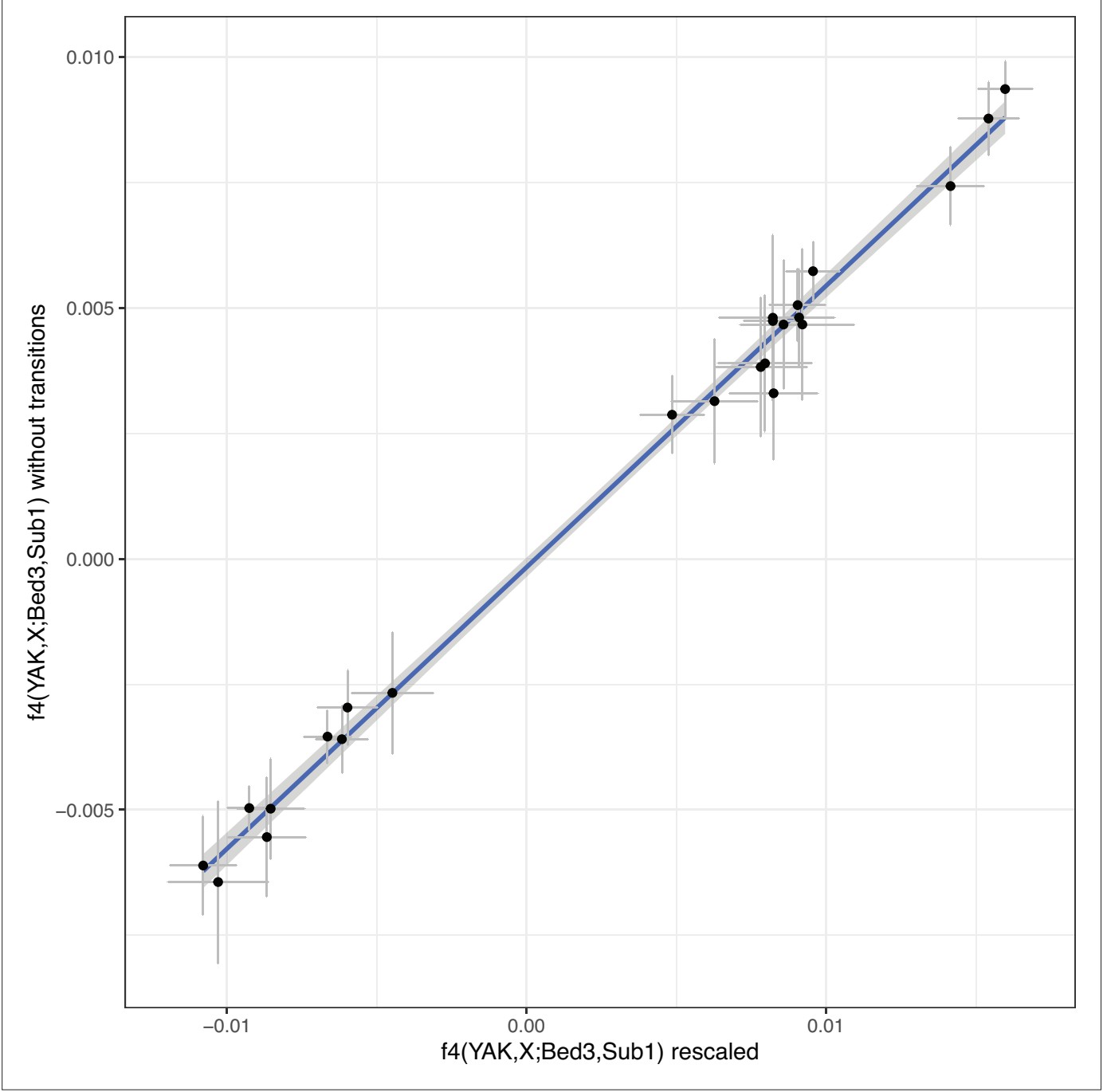

**Figure 5.** $f_4$ statistics contrasting our newly sequenced samples to a reference domestic individual and a reference European aurochs. The $f_4$ statistics were calculated for two different versions of the bam files: with rescaled bases in fragment ends vs untreated bam files but an SNP panel excluding transitions. The blue line indicates a linear regression with confidence interval. Error bars indicate 95% confidence intervals estimated using a block jackknife procedure.

also used Struct-f4 (*Librado and Orlando, 2022*) to estimate ancestry proportions. First, input files were generated with the provided helper scripts and $f_4$ statistics were calculated in blocks of 5Mbp. Struct-f4 was then run in semi-supervised mode with default parameters to estimate ancestries in Iberian individuals with at least 0.1× coverage. This cutoff was chosen as lower coverage samples prevented conversion. CPC98, YAK, Ch22, Gyu2, Bed3, Sub1, and Sha_3b were used as additional individuals to provide a framework of different possible ancestries.

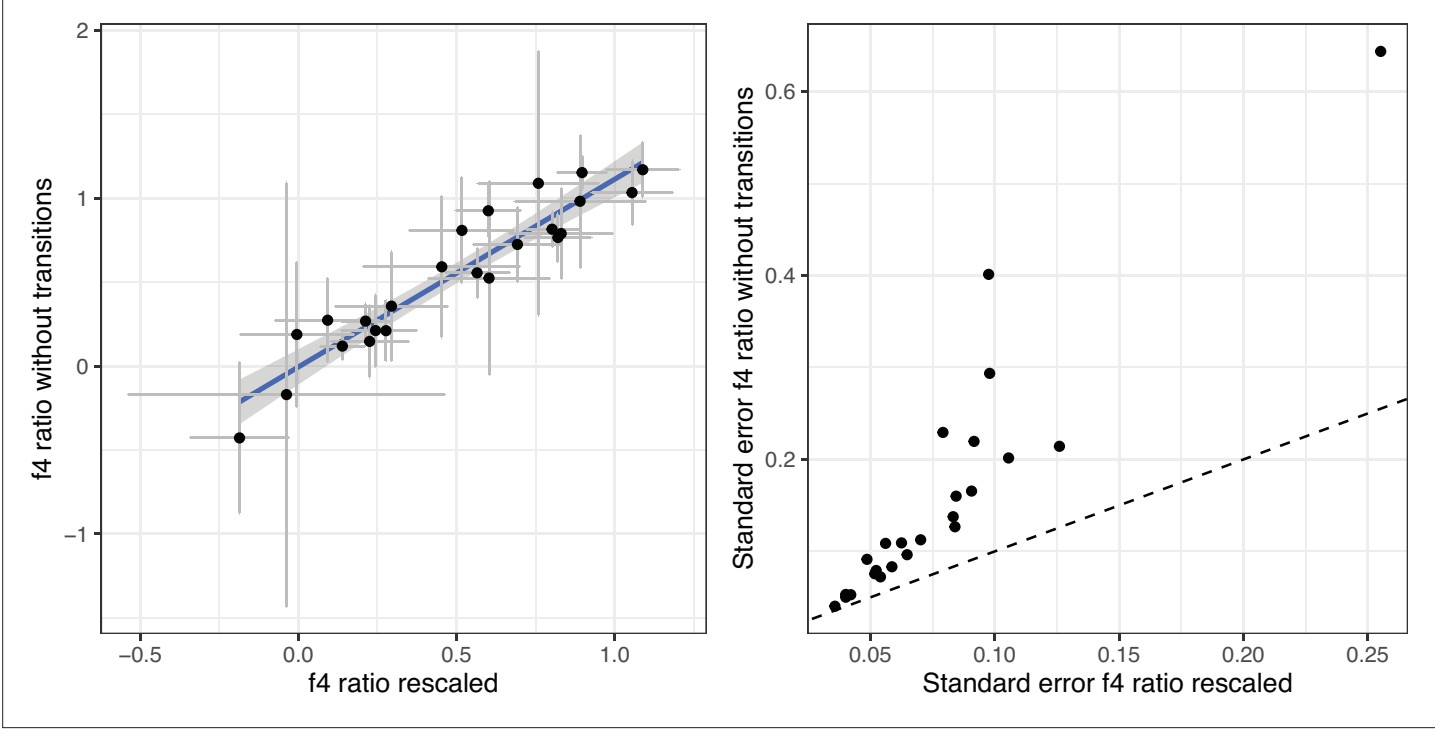

**Figure 6.** $f_4$ ratios estimating European aurochs ancestry in our newly sequenced Iberian samples. The $f_4$ ratios were calculated for two different versions of the bam files: with rescaled bases in fragment ends vs untreated bam files but an SNP panel excluding transitions. The right panel shows the standard errors for the $f_4$ ratio estimates, the dashed line would correspond to equal standard errors for the two panels. Error bars indicate 95% confidence intervals estimated using a block jackknife procedure.

## Acknowledgements

We are incredibly grateful to the excavation teams at the four archaeological sites. Sequencing was performed at The National Genomics Infrastructure (NGI) Stockholm. The computations and data handling were enabled by resources provided by the National Academic Infrastructure for Supercomputing in Sweden (NAISS) and the Swedish National Infrastructure for Computing (SNIC) at Uppmax, partially funded by the Swedish Research Council through grant agreements no. 2022-06725 and no. 2018-05973. This project was supported by grants from the Royal Physiographic Society of Lund (Nilsson-Ehle Endowments) to TG and CV, Vetenskapsrådet (2017-05267) to TG, Ramón y Cajal (RYC2018-025223-I) to CV and a Beatriz Galindo Fellowship (BGS220-461AA-69201) to CS. The Atapuerca research project is financed by the Ministerio de Ciencia, Innovación y Universidades grant PID2021-122355NB-C31 funded by the MCIN/AEI/10.13039/501100011033 and ERDF 'A way of making Europe'.

## Additional information

### Funding

| Funder | Grant reference number | Author |
| --- | --- | --- |
| Vetenskapsrådet | 2017-05267 | Torsten Günther |
| Royal Physiographic Society of Lund | | Torsten Günther Cristina Valdiosera |
| Ramón y Cajal | RYC2018-025223-I | Cristina Valdiosera |
| Beatriz Galindo Fellowship | BGS220-461AA-69201 | Colin Smith |

| Funder | Grant reference number | Author |
|---|---|---|
| Ministerio de Ciencia, Innovación y Universidades | PID2021-122355NB-C31 | Eneko Iriarte<br>Rebeca García-González<br>Amalia Pérez<br>Juan Luis Arsuaga<br>José-Miguel Carretero<br>Colin Smith<br>Cristina Valdiosera |
| European Regional Development Fund | A way of making Europe | Eneko Iriarte<br>Rebeca García-González<br>Amalia Pérez<br>Juan Luis Arsuaga<br>José-Miguel Carretero<br>Colin Smith<br>Cristina Valdiosera |

The funders had no role in study design, data collection and interpretation, or the decision to submit the work for publication.

## Author contributions

Torsten Günther, Conceptualization, Resources, Data curation, Software, Formal analysis, Supervision, Funding acquisition, Validation, Investigation, Visualization, Methodology, Writing – original draft, Project administration, Writing – review and editing; Jacob Chisausky, Formal analysis, Investigation, Writing – original draft, Writing – review and editing; Ángeles M Galindo-Pellicena, Resources, Data curation, Formal analysis, Writing – review and editing; Eneko Iriarte, Resources, Investigation, Methodology, Writing – review and editing; Oscar Cortes Gardyn, Resources, Data curation, Validation, Investigation, Writing – review and editing; Paulina G Eusebi, Alfonso Alday, Manuel Rojo, Resources, Data curation, Investigation, Writing – review and editing; Rebeca García-González, Marta Moreno-García, Cristina Tejedor Rodríguez, Resources, Data curation, Writing – review and editing; Irene Ureña, Data curation, Writing – review and editing; Amalia Pérez, Iñigo García Martínez de Lagrán, Resources, Writing – review and editing; Juan Luis Arsuaga, Resources, Funding acquisition, Project administration, Writing – review and editing; José-Miguel Carretero, Resources, Data curation, Supervision, Funding acquisition, Project administration, Writing – review and editing; Anders Götherström, Resources, Supervision, Funding acquisition, Project administration, Writing – review and editing; Colin Smith, Cristina Valdiosera, Conceptualization, Resources, Data curation, Formal analysis, Supervision, Funding acquisition, Validation, Investigation, Visualization, Methodology, Writing – original draft, Project administration, Writing – review and editing

## Author ORCIDs

Torsten Günther (iD) https://orcid.org/0000-0001-9460-390X
Oscar Cortes Gardyn (iD) https://orcid.org/0000-0001-7685-3980
Marta Moreno-García (iD) https://orcid.org/0000-0002-6735-9355
Cristina Valdiosera (iD) https://orcid.org/0000-0003-4948-2226

Reviewer #2 (Public review): https://doi.org/10.7554/eLife.93076.3.sa1
Reviewer #3 (Public review): https://doi.org/10.7554/eLife.93076.3.sa2
Author response https://doi.org/10.7554/eLife.93076.3.sa3

# Additional files

## Supplementary files

Supplementary file 1. Excel sheet containing three tables: (i) sample list, (ii) ancestry estimates, and (iii) metric and isotope data.

MDAR checklist

## Data availability

Raw sequence data and aligned reads for the new ancient individuals are available through the European Nucleotide Archive under accession number PRJEB63140. Pseudohaploid genotype calls can be obtained from Zenodo. All metric and isotope data are available in *Supplementary file 1*.

The following datasets were generated:

| Author(s) | Year | Dataset title | Dataset URL | Database and Identifier |
|---|---|---|---|---|
| Günther T | 2024 | Sequence data | https://www.ebi.ac.uk/ena/browser/view/PRJEB63140 | European Nucleotide Archive, PRJEB63140 |
| Günther T, Valdiosera C | 2025 | SNP calls for ancient Iberian cattle and aurochs | https://doi.org/10.5281/zenodo.14652391 | Zenodo, 10.5281/zenodo.14652390 |

The following previously published datasets were used:

| Author(s) | Year | Dataset title | Dataset URL | Database and Identifier |
|---|---|---|---|---|
| Eusebi PG, Cort SS, Contreras E, Dunner S, Sevane N | 2022 | Lidia genome sequences | https://www.ncbi.nlm.nih.gov/sra/?term=PRJNA838078 | NCBI Sequence Read Archive, PRJNA838078 |
| Verdugo MP, Mullin VE, Scheu A, Mattiangeli V, Daly KG, Delser PM, Hare AJ, Burger J, Collins MJ, Kehati R | 2019 | Ancient cattle and aurochs sequences | https://www.ebi.ac.uk/ena/browser/view/PRJEB31621 | ArrayExpress, PRJEB31621 |
| Erven JAM, Scheu A, Verdugo MP, Cassidy L, Chen N, Gehlen B, Street M, Madsen O, Mullin VE | 2024 | High coverage central European aurochs sequences | https://www.ebi.ac.uk/ena/browser/view/PRJEB74338 | ArrayExpress, PRJEB74338 |
| Stephen DEP, David AM, McGettigan PA, Matthew DT, Ceiridwen JE, Amanda JL, Alison M, Martin B, Mark TD, Yuan L, Andrew TC, Albrecht VR, Steven S, Charles S, Shuaishuai T, Daniel GB, Tad SS, Brendan JL, MacHugh DE | 2015 | Bos primigenius isolate:CPC98 Raw sequence reads | https://www.ncbi.nlm.nih.gov/bioproject/PRJNA294709/ | NCBI BioProject, PRJNA294709 |
| Naval-Sanchez M, Porto-Neto L, Daetwyler H, Hayes B, Reverter-Gomez T | 2019 | Allele frequencies between Bos taurus and Bos indicus | https://doi.org/10.25919/5ceb24e4ae2f8 | CSIRO Data Access, 10.25919/5ceb24e4ae2f8 |
| Upadhyay MR, Chen W, Lenstra JA, et al | 2016 | Data from: Genetic origin, admixture and population history of aurochs (Bos primigenius) and primitive European cattle | https://doi.org/10.5061/dryad.f2d1q | Dryad Digital Repository, 10.5061/dryad.f2d1q |

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

# Appendix 1

## Site description

### Artusia

The Artusia rock shelter is located at Unzué, in the eastern part of Navarre (Spain). It is positioned at the southern slope of a narrow path excavated between the Alaiz Mountains on the north, and the Unzué Peak on the south, by the Artusia stream. Its geographical location is characterised by a natural communication route between the aforementioned mountains and the Ebro valley, and by its proximity to differentiated biotopes and landscapes, similar to other Mesolithic shelters in the Ebro valley.

The rock shelter is filled by a stratigraphic package cut lengthwise by the Artusia stream exposing the different archaeological levels. Consequently, the archaeological works carried out in the shelter was a rescue excavation to prevent the collapse of the stratigraphy. Five phases of occupation were defined during the excavations, all of them within the Mesolithic, represented by the same technological evolution defined in the Ebro valley for this period:

- Artusia I: First phase of the Mesolithic of Notches and Denticulates: 7461–7145 cal BC: Abundant faunal and flint remains were recovered together with charcoal remains derived from successive human occupations.
- Artusia II: Second phase of the Mesolithic of Notches and Denticulates: 6689–6507 cal BC: The stratigraphic units show recurring human occupations of seasonal character. The presence of occupation floors and hearths and the materials recovered (large quantities of flint and faunal remains) indicate that these human occupations were very intense.
- Artusia III: First phase of the Geometric Mesolithic: 6598–6453 cal BC: In general, the characteristics of the occupation layers and the hearths are the same. However, in Artusia III the occupations seem to be less intense than in Artusia II.
- Artusia IV: Second phase of the Geometric Mesolithic. There are no diagnostic materials, just charcoal and fauna remains.
- Artusia V: Third phase of the Geometric Mesolithic: 6205–6009 cal BC: Artusia V presents occupational contexts and hearths similar to those of previous phases.

In addition, Artusia stands out for the paleoclimatic information of its stratigraphy. Two different climatic events are detected: the 8.5 ka BP event (ca. 6550–6450 cal BC) and the 8.2 ka BP climate event (ca. 6300–6140 cal BC). These events were cold and dry and had important consequences for the landscapes and for the human groups. In this context, Mesolithic communities most likely suffered a scarcity of prey in an increasingly open environment (aridity and deforestation), which potentially required hunting techniques to be adapted in order to suit this new environment, e.g., projectile points made using geometric microliths. Altogether, the archaeological and environmental analysis of the Artusia rockshelter reinforces the idea that these hunter-gatherer groups had to adapt their lithic technologies and their lifestyles and subsistence strategies to a changing environment during the last phase of the Mesolithic.

### Els Trocs

Els Trocs Cave (San feliu de Veri/Bisaurri, Huesca) is located at an over 1500 m altitude in a high mountain environment in the Pyrenees of Huesca, in the region of Alta Ribagorza in Aragon. Specifically, it is located on the southern slope of a conical hill, very close to the town of San Feliu de Veri (Municipality of Bisaurri, Huesca), and next to a plateau called partida de la Selvaplana, to the north of the Turbón massif and close to the Axial Pyrenees, equidistant from the rivers Ésera and Isábena. The site location is a strategic point where transhumance paths (cabañeras) that have almost certainly been used since the Neolithic converge. The site stands out for its proximity to fresh high altitude pastures, as well as to two saline springs, one of them, La Muria, less than 1 km from the cave, rich in salts such as chlorine and sodium, essential for sheep feeding due to their absence from the herbaceous pasture.

Despite the difficult conditions of temperature and humidity of the cavity, a complex unaltered stratigraphy has been described documenting a sequence of human occupation throughout two millennia. Based on the stratigraphic sequence and a set of radiocarbon dates, up to five moments of occupation of the cave can be identified:

- Trocs I: This is the first phase of occupation of the site, dated to the beginning of the last third of the sixth millennium cal BC. The bovine remains analysed in this work belong to this cycle of use of the cave.
- Trocs II: Phase dated to the middle of the fifth millennium cal BC characterised, above all, by the presence of important amortised hearths.
- Trocs III: Uniform stratigraphic horizon and of notable power that is formed along almost a millennium; between the first third of the fourth millennium, until the beginnings of the third millennium cal BC. The presence of numerous smaller households than the previous ones reinforces the recurrent occupations of very short duration.
- Trocs IV: Partial funerary deposit from the end of the Neolithic and beginning of the Chalcolithic (third millennium cal BC).
- Trocs V: The last occupation of the cave corresponds to historical times and is the most superficial phase of the stratigraphy, where materials from the Imperial Roman period have been found, including remains of ceramics, glass, iron objects, and two coins that would allow a dating in the Lower Empire (fourth century AD).

The dating of the different occupational levels of the Els Trocs cave is supported by a series of radiocarbon dates that have been carried out for each phase of the site. These dates, based on samples or short-lived events (cereal seeds, domestic fauna, or human bones), have allowed us to determine the chronology of the site (*Tejedor-Rodríguez et al., 2021*; *Rojo et al., 2015*; *Rojo-Guerra et al., 2013*; *Rojo-Guerra et al., 2018*).

## El Portalón de Cueva Mayor

This archaeological site is located in the karstic system of the Sierra de Atapuerca in Burgos, northern Spain. Archaeological excavations at El Portalón cave have exposed a stratigraphic sequence comprising the Late Pleistocene and the Holocene. A detailed record of more than 90 radiocarbon dates for the entire stratigraphic sequence range from 30,000 BP to 1000 BP (*Carretero et al., 2008*; *Pérez-Romero et al., 2017*; *Pérez-Romero, 2021*). While the Late Pleistocene sediments show scarce archaeological activity, the Holocene period is characterised by intense human occupation from the Mesolithic to Medieval times (*Pérez-Romero, 2021*). All chronocultural periods are represented by abundant archaeological artefacts such as stone tools, ceramics, worked bone, etc. Human remains are present throughout the whole period, but domestic faunal remains are much more frequent. Ovicaprines, cattle, and pigs are the three most abundant domestic species, respectively, for the Neolithic, Chalcolithic, and Early Bronze Age, but this pattern changed by the Middle Bronze Age where cattle became more abundant than ovicaprines (*Francés-Negro et al., 2021*; *Galindo-Pellicena et al., 2020*). Aurochs (*B. primigenius*) remains have also been identified at the site all throughout the Neolithic, Chalcolithic, and Bronze Age (*Galindo-Pellicena et al., 2020*). The archaeological evidence suggests that most of these domestic animals were used for human consumption (*Francés-Negro et al., 2021*), however, these have also been found as grave goods in a child burial during the Chalcolithic (*Pérez-Romero et al., 2017*).

## Mendandia

The prehistoric deposit of Mendandia was discovered in 1991 in the Upper Ebro River Basin, in the town of Sáseta (Treviño, Spain). It was discovered and excavated by A Alday between 1992 and 1995 and in 1997: work was carried out on 13 m², which made it possible to differentiate a stratigraphy with five sedimentological levels and six cultural units. It is a continuous sequence, without erosive phases, whose characters derive from natural sedimentation and human activity, providing very dense archaeological collections (*Alday, 2006*). It is a rock shelter of medium dimensions (15 m long by 5 m wide) settled on a wide terrace of 27 m by 14 m. At its feet runs the river Ayuda, the great collector of the Treviño Basin. The catchment area is located on the Oquina-Sáseta Ravine, a natural north-south corridor, surrounded by valley, mid-mountain, and even high altitude environments with peaks close to a thousand metres in altitude. The plant and animal resources available during prehistoric times were abundant and diverse, making it an ideal human settlement for several generations: 2000 years of uninterrupted visitation have been recorded.

Five lithostratigraphic horizons (from bottom to top V, IV, III, II, and I) and six cultural stages have been described. The observations and different granulometric, mineralogical, and chemical analyses carried out on the whole sequence and for well-chosen samples show a strong anthropic character to the fill (contributions of hunted fauna, tools, prepared fires).

The first occupations (Level V) date to 8500±60 BP (GrA-6874), and correspond to groups of small size and not very intense activities. Based on the characters and composition of the lithic instruments the complex is ascribed to the Mesolithic of laminar facies. The site was used for hunting horses, aurochs, goats, sheep, deer, roe deer, and wild boar. Level IV is also Mesolithic but of the notched and denticulate facies with a chronology between 7810±50 (GrN-22744) and 7780±40 BP (GrN-22745). An unusually high number of 47,579 bone fragments were inventoried for this type of site: deer and roe deer were objects of special interest to the human group, but the importance of the aurochs is very striking, with 80% of them being less than 2 months of age, which indicates a very specific hunting management strategy. The lithic industry includes 11,284 items, with 354 retouched objects: 35 scrapers, 58 perforators, more than a hundred and a half very characteristic notches and denticulates, 9 burins, among other retouched objects. The Mesolithic cycle ends with the last of the units, the geometric, present in Mendandia in level III-lower. Chronologically, the substitution of the campiñoid forms by the new microlithic ones is very fast moving into 7620±50 BP (GrN-22743). The Mesolithic cycle is replaced by the Neolithic, represented in Mendandia by levels III-superior, II and I. The antiquity of the horizons with dates of 7210±80 (GrN-19658) and 7180±45 BP (GrN-22742) (both for III-superior) and 6540±70 (GrN-22741) and 6440±40 BP (GrN-22740) for II and I, respectively, is undoubtedly surprising. Although during the Neolithic the technological preferences vary, the use of the site does not change. The remains of fauna (horse, aurochs, goat, buck, deer, roe deer, wolf, fox, marten, and rabbit) are still very numerous. There is no evidence of domestication. The occupation of the site ends after the early Neolithic with no further occupations.

Overall, almost 50,000 bone remains have been recovered, including horses, aurochs, goats, bucks, deer, roe deer, and wild boar, as major species, but also wolves, foxes, and martens. Based on the ages and sexes of the trapped animals, the hypothesis of a recurrent nomadism can be accepted: they would go to the refuge between late spring and early summer to harass mainly young and females, in a strategy with control of the hunted in order not to decimate the resources.

Hunting occupied much of the time of the inhabitants of Mendandia. The strategic location of the shelter enabled the groups to capture a wide variety of species: with certain variations in their percentages, 90% of the game hunted was roe deer, red deer, and aurochs, followed, more distantly, by wild boar, goats, sheep, horses, foxes, wolves, badgers, and wild cats, among others. The analysis of the age of the animals allows us to consider spring and early summer as the phase of greatest hunting activity. Moreover, the proportion of anatomical parts present and absent from each animal suggests that, in many cases, the meatier parts of the animals were consumed elsewhere by means of conservation practices such as smoking. The anthracological study would indicate the burning of wet wood for precisely this purpose.

# Appendix 2

## Data analysis

### DNA data authentication

All samples included in the nuclear analysis show deamination patterns characteristic for authentic aDNA (*Appendix 2—figures 1–3*).

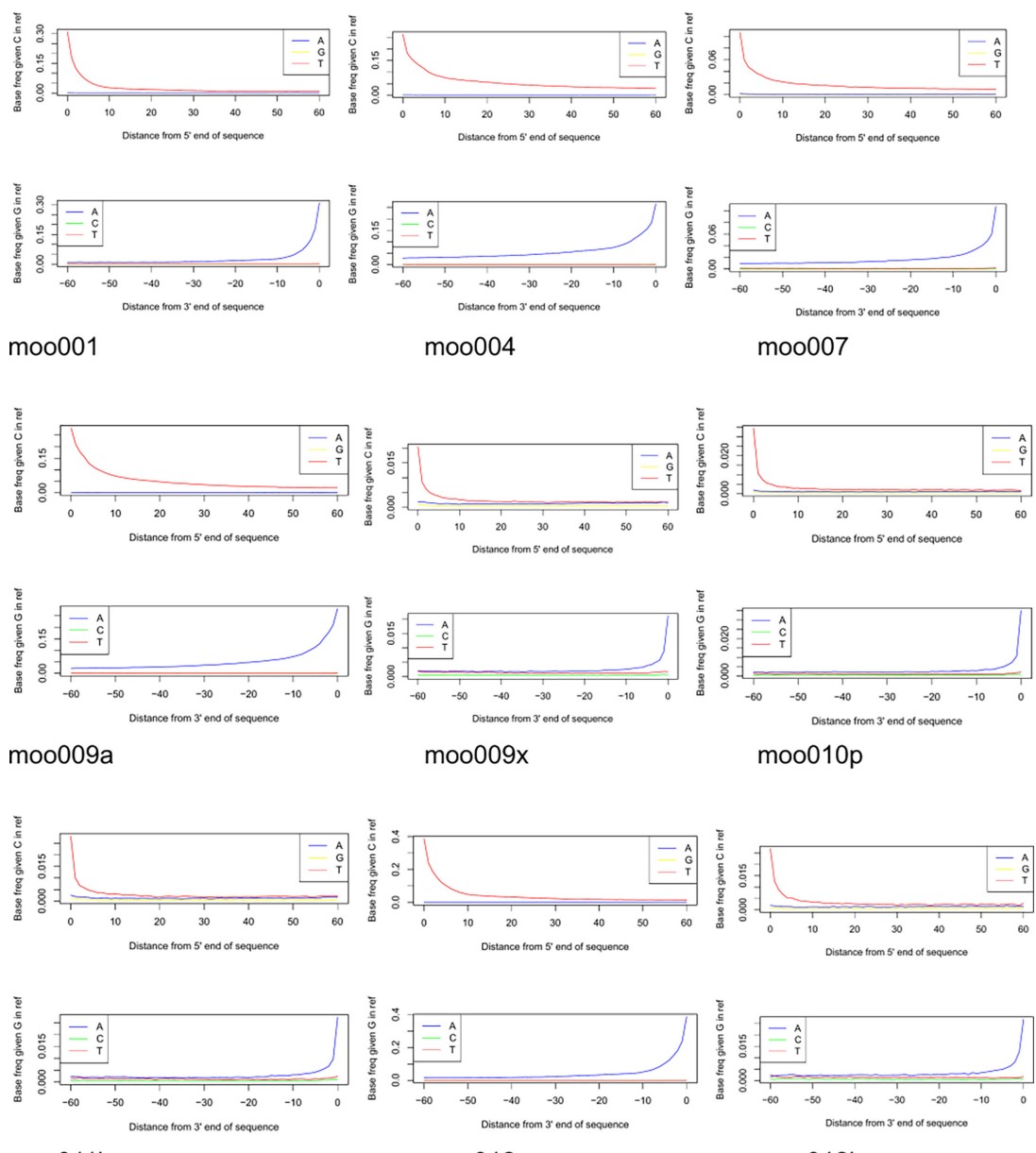

**Appendix 2—figure 1.** Misincorporation plots for the samples moo001–moo012b included in nuclear analysis.

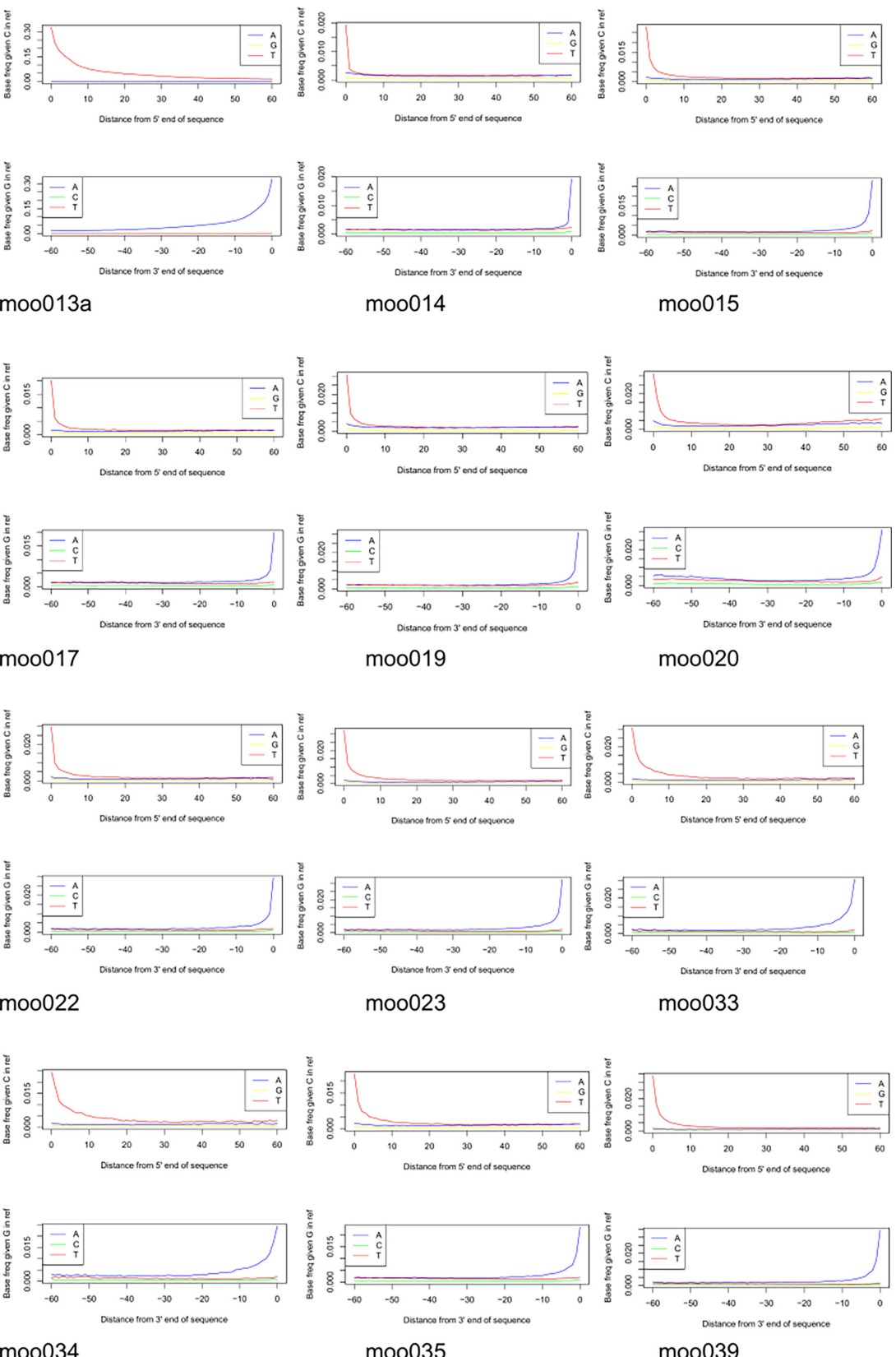

**Appendix 2—figure 2.** Misincorporation plots for the samples moo013a–moo039 included in nuclear analysis.

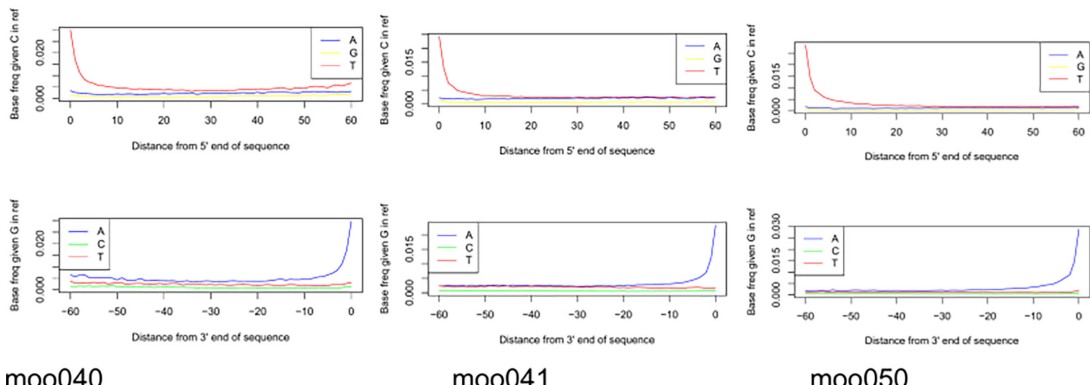

**Appendix 2—figure 3.** Misincorporation plots for the samples moo040–moo050 included in nuclear analysis.

## Principal component analysis

A PCA was conducted for the ancient samples with at least 0.01× coverage and modern breeds from *Upadhyay et al., 2017*. We used smartpca (*Patterson et al., 2006*) with numoutlier: 0, killr2: YES, r2thresh: 0.4, lsqproject: YES, and shrinkmode: YES to project the ancient samples onto the genetic variation defined by the modern breeds. Results are shown in *Appendix 2—figures 4 and 5*. All Iberian samples fall close to the centre of the plot, near modern central European breeds and the British aurochs CPC98. Some later samples appear to show stronger affinities to modern Iberian breeds. However, we do not see clear separation between predominantly aurochs or predominantly domestic individuals based on this analysis, highlighting the limitations of a PCA based on modern genetic variation if this is only a subset of the ancient variation that was present in the extinct wild ancestor.

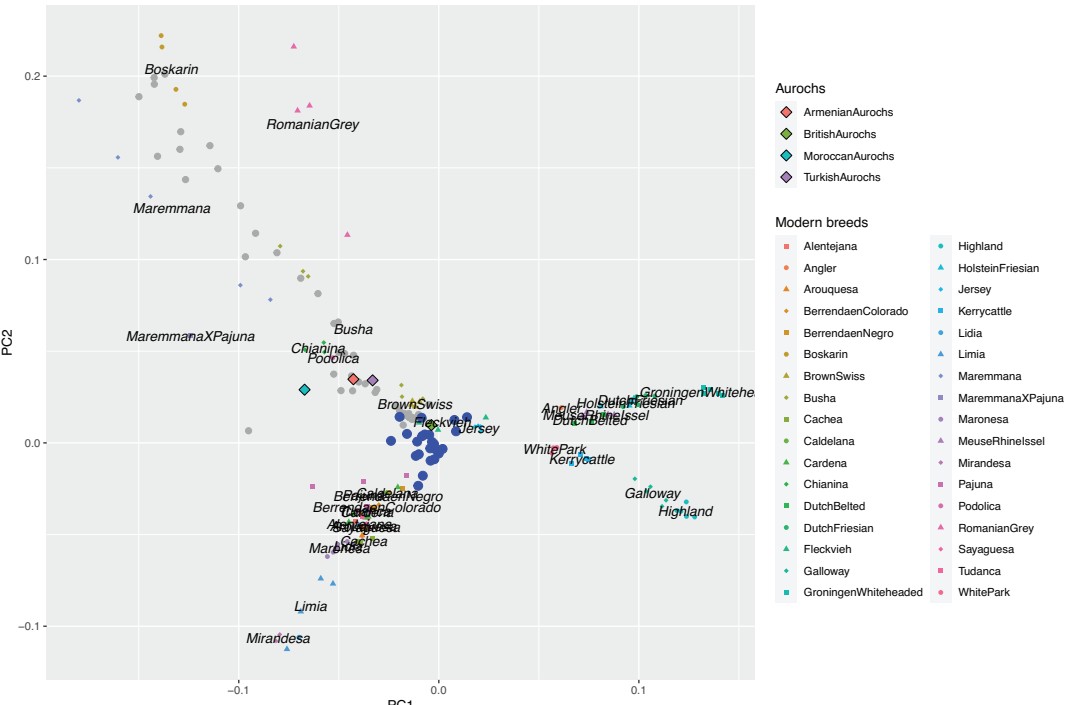

**Appendix 2—figure 4.** Biplot of PC1 and PC2 defined by modern western breeds together with ancient samples projected onto the PC-space. Aurochs genomes are shown as diamonds, ancient Iberian samples as blue dots, and other ancient samples from *Verdugo et al., 2019*, as grey dots.

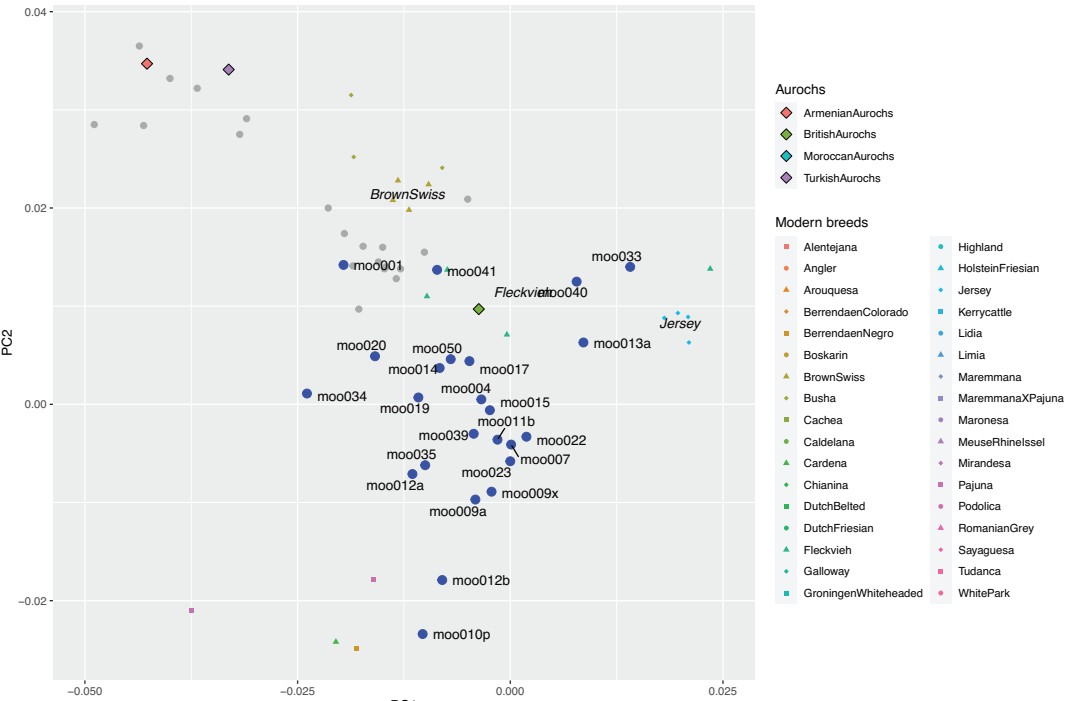

**Appendix 2—figure 5.** Zoomed version of a biplot of PC1 and PC2 defined by modern western breeds together with ancient samples projected onto the PC-space. Aurochs genomes are shown as diamonds, ancient Iberian samples as blue dots, and other ancient samples from *Verdugo et al., 2019*, as grey dots. Sample IDs are added for Iberian samples.

## Model-based clustering

Unsupervised clustering of the ancient individuals together with modern breeds was carried out using ADMIXTURE (*Alexander et al., 2009*; *Alexander and Lange, 2011*). All individuals were haploidised by randomly drawing a single allele at each site to avoid artificial drift in the pseudohaploid ancient individuals. We then used plink (*Purcell et al., 2007*; *Chang et al., 2015*) and the parameter `--indep-pairwise` 200 25 0.4 for linkage disequilibrium pruning. ADMIXTURE was run from K=2 to K=10 with 20 different seeds per K. Representative runs were then identified using pong in greedy mode with a similarity threshold of 0.95 (*Behr et al., 2016*). Results are displayed in *Appendix 2—figure 6*. Similar to the PCA, the modern genetic variation dominates the cluster assignment not allowing for a differentiation between European aurochs, ancient European domestics, and hybrid individuals. For higher numbers of clusters, there appear some differences in certain ancestry proportions between Iberian samples that are assumed to represent aurochs or domestic cattle (see other analysis) but all samples still carry ancestry from the same clusters and no cluster is exclusively representing European aurochs populations. This is likely a consequence of the dataset being heavily biased towards modern breeds which only represent a subset of the extinct aurochs variation.

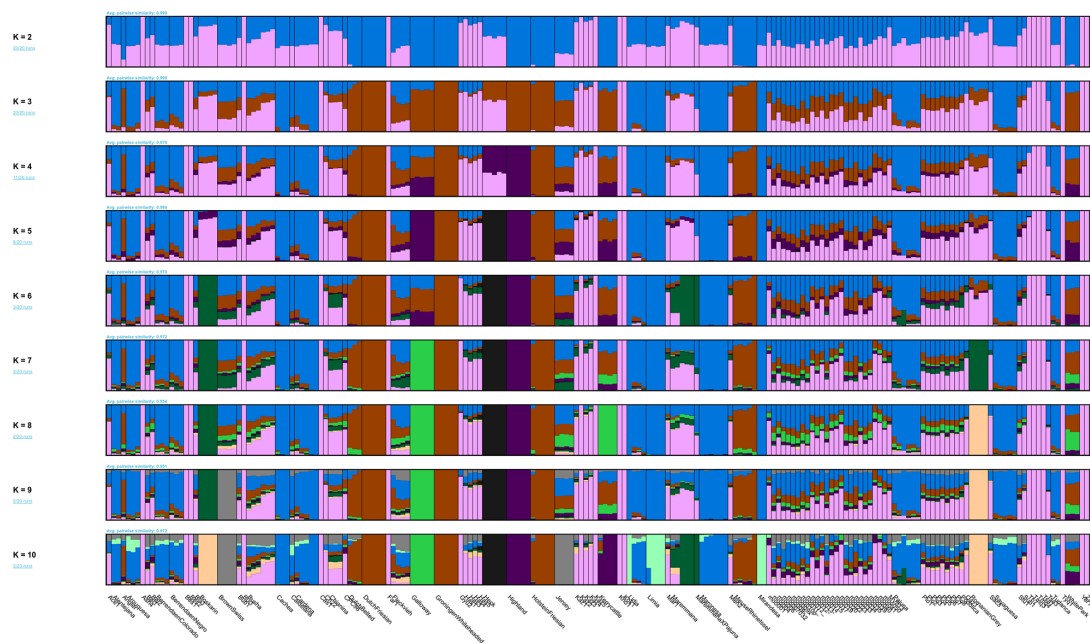

**Appendix 2—figure 6.** Model-based clustering results for modern breeds and ancient *Bos* samples from K=2 to K=10.

## Ancestry modelling in ancient Iberian samples

In addition to qpAdm and f₄ ratios, we also used Struct-f4 (*Librado and Orlando, 2022*) to estimate ancestry proportions in the ancient Iberian samples. We selected only ancient Iberian samples with at least 0.1× coverage and opted for a semi-supervised approach with the European aurochses Bed3 and CPC98, as well as YAK, Sha_3b (zebu cattle), Sub1 (Anatolian Neolithic), Gyu2 (Caucasus aurochs), and Ch22 (Anatolian aurochs) as reference samples.

At K=2, the two outgroups (Yak and indicine cattle) form a separate group (*Appendix 2—figure 7*), at K=3 each of them receive their own cluster (*Appendix 2—figure 8*) without any admixture in the Iberian samples. At K=4, the European aurochs (Bed3 and CPC98) separates from the domestic samples and western Asian aurochs. All tested Iberian samples now display non-zero proportions of both of these ancestries (*Appendix 2—figure 9*). At K=5, the Caucasian aurochs (Gyu2) splits into its own cluster with some Iberian samples displaying small proportions of this ancestry in addition to the two other components (*Appendix 2—figure 10*).

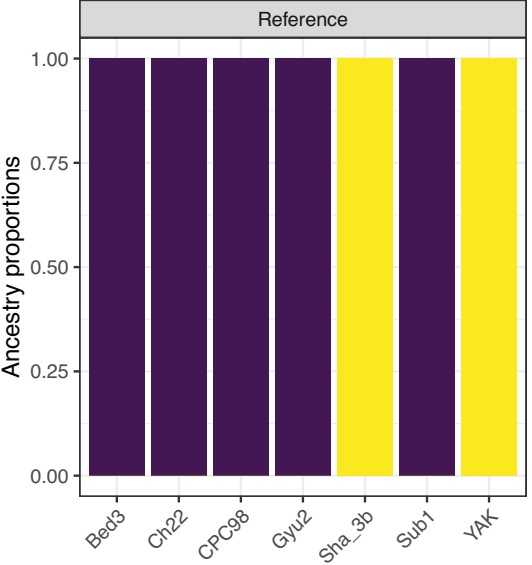
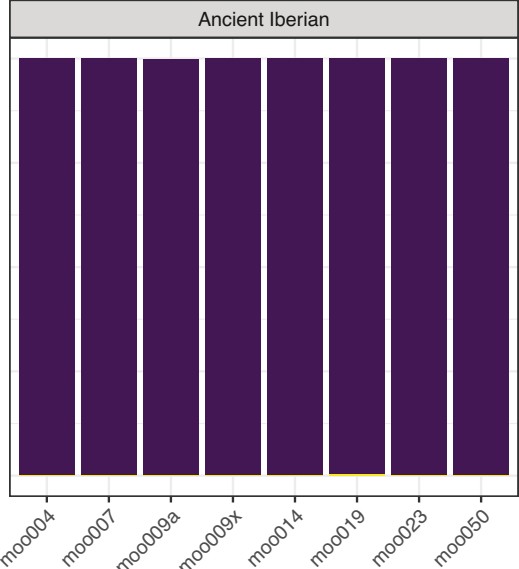

**Appendix 2—figure 7.** Structf4 results for K=2.

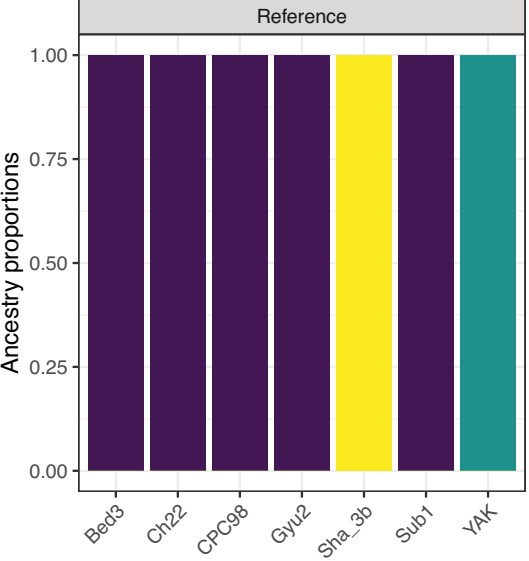
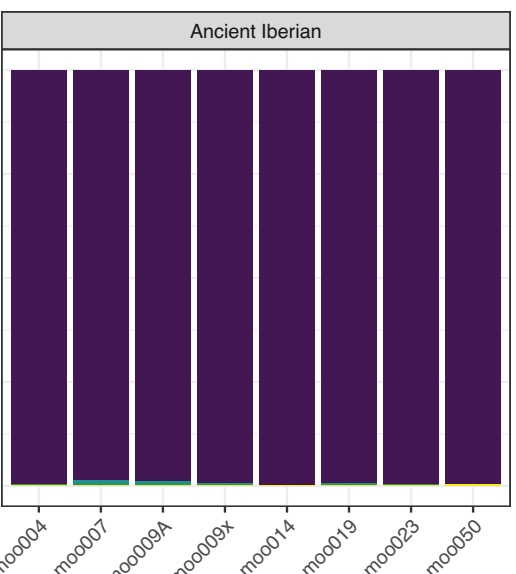

**Appendix 2—figure 8.** Structf4 results for K=3.

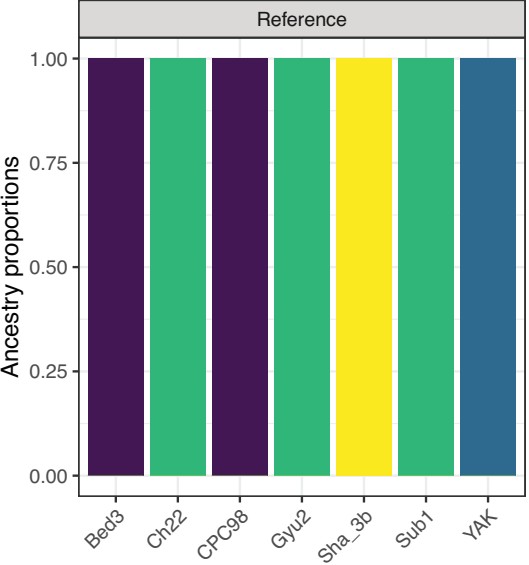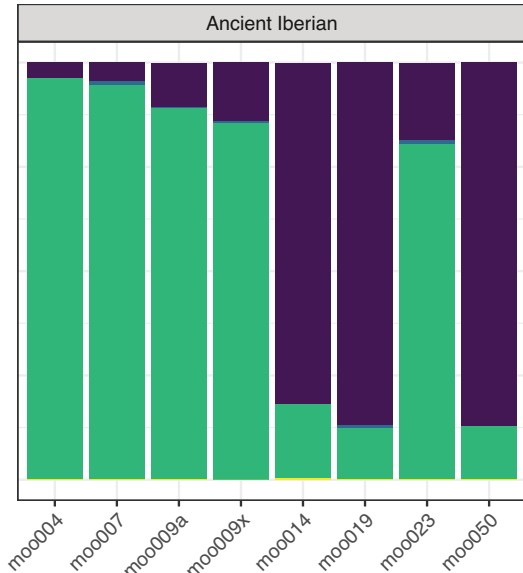

**Appendix 2—figure 9.** Structf4 results for K=4.

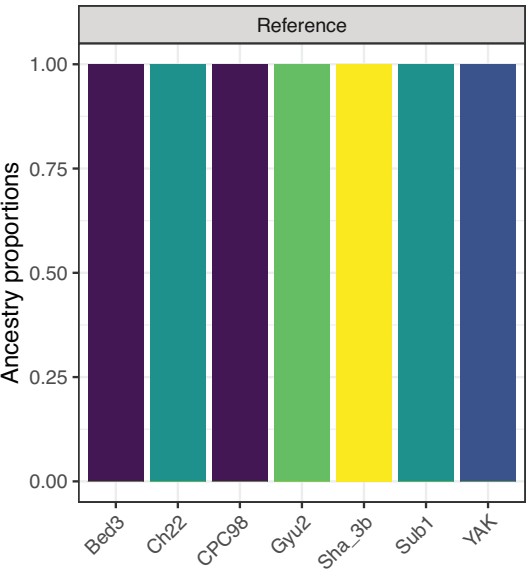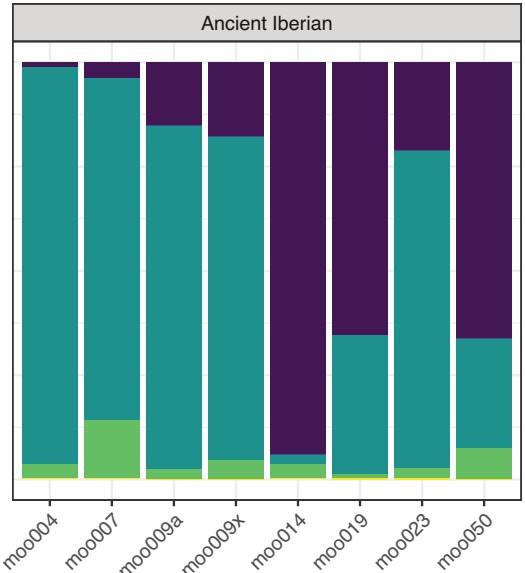

**Appendix 2—figure 10.** Structf4 results for K=5.

## Ancestry modelling in modern breeds

Since the publication of the first European aurochs genome, the Mesolithic British aurochs (CPC98) (*Park et al., 2015*), several studies have contrasted different modern breeds in their amount of allele sharing with the aurochs genome (*Park et al., 2015*; *Upadhyay et al., 2017*; *da Fonseca et al., 2019*) using *D* statistics (*Durand et al., 2011*). Such *D* statistics provide an assessment of relative allele sharing of two focal breeds with an outgroup but they do not directly estimate the amount of aurochs ancestry in each breed. The currently available data allows us to use more advanced methods that estimate the proportion of ancestries and are also able to reject certain ancestry models. Similar to the analysis of ancient samples, we used qpAdm (*Haak et al., 2015*; *Harney et al., 2021*) to model 30 modern European breeds with ancestry from two sources: Bed3 and Neolithic domestic Anatolian (Sub1). We only used Western and Central European breeds as target populations as Southern and Eastern European breeds showed signals of zebu cattle admixture (*Upadhyay et al., 2017*) which would violate our two-source model. Proportions of estimated aurochs ancestry are relatively similar

across all breeds, ranging from 20.9% (Pajuna) to 28.1% (Dutch Friesian), see *Appendix 2—table 1*. It is important to note that the standard errors of these estimates are all around 3%, so the 95% confidence intervals would be overlapping and the differences between breeds are not significant. The two-source model was sufficient to explain the ancestry in 25 of the tested breeds (p>0.01) while the model was rejected for 5 breeds (p<0.01) including the commercial Holstein Friesian suggesting that they may contain small proportions of ancestries from outside of Europe. Estimates for the six individual Lidia genomes (*Eusebi et al., 2022*) are shown in *Figure 4—figure supplement 1*.

**Appendix 2—table 1.** qpAdm modelling of the genomic ancestry in modern European cattle breeds under a two-source model using European aurochs (Bed3) and Anatolian Neolithic cattle (Sub1) as representatives of the sources.

Genotypes for modern breeds from *Upadhyay et al., 2017* (C=commercial breed).

| Breed code | Breed | Origin | Proportion European aurochs | Proportion Anatolia Neolithic | Standard error | p-Value |
|---|---|---|---|---|---|---|
| AL01 | Alentejana | Portugal | 0.2502 | 0.7498 | 0.0361 | 0.1030 |
| AN01 | Angler (C) | Germany | 0.2746 | 0.7254 | 0.0401 | 0.0747 |
| AR01 | Arouquesa | Portugal | 0.2572 | 0.7428 | 0.0341 | 0.0555 |
| BC01 | Berrenda en colorado | Spain | 0.2738 | 0.7262 | 0.0354 | *0.0077* |
| BN01 | Berrenda en negro | Spain | 0.3005 | 0.6995 | 0.0343 | *0.0014* |
| BS01 | Brown Swiss (C) | Switzerland | 0.2480 | 0.7520 | 0.0355 | 0.0343 |
| CA01 | Cardena | Spain | 0.2245 | 0.7755 | 0.0338 | 0.0141 |
| CC01 | Cachena | Portugal | 0.2738 | 0.7262 | 0.0372 | 0.0564 |
| CL01 | Caldelana | Spain | 0.2540 | 0.7460 | 0.0388 | 0.3139 |
| DB01 | Dutch Belted (C) | The Netherlands | 0.2724 | 0.7276 | 0.0358 | 0.2135 |
| DF01 | Dutch Friesian (C) | The Netherlands | 0.2814 | 0.7186 | 0.0364 | 0.0413 |
| FL01 | Fleckvieh (C) | Switzerland | 0.2801 | 0.7199 | 0.0335 | 0.0685 |
| GA01 | Galloway | Scotland | 0.2357 | 0.7643 | 0.0350 | 0.4203 |
| GW01 | Groningen Whiteheaded (C) | The Netherlands | 0.2479 | 0.7521 | 0.0353 | 0.3394 |
| HF01 | Holstein Friesian (C) | The Netherlands | 0.3402 | 0.6598 | 0.0352 | *0.0068* |
| HL01 | Highland | Scotland | 0.2323 | 0.7677 | 0.0375 | 0.0788 |
| JE01 | Jersey (C) | Jersey Island | 0.2732 | 0.7268 | 0.0352 | 0.3093 |
| KC01 | Kerry Cattle | Ireland | 0.2416 | 0.7584 | 0.0357 | 0.1596 |
| LI01 | Lidia | Spain | 0.2502 | 0.7498 | 0.0333 | 0.2204 |
| LM01 | Limia | Spain | 0.2337 | 0.7663 | 0.0347 | 0.3224 |
| ME01 | Maronesa | Spain | 0.2668 | 0.7332 | 0.0351 | 0.0339 |
| MI01 | Mirandesa | Portugal | 0.2643 | 0.7357 | 0.0391 | 0.0942 |
| MN01 | Maronesa | Spain | 0.2718 | 0.7282 | 0.0361 | 0.0400 |
| MR01 | MRY (C) | The Netherlands | 0.2564 | 0.7436 | 0.0344 | 0.0769 |
| PA01 | Pajuna | Spain | 0.2415 | 0.7585 | 0.0327 | *0.0097* |
| PA02 | Pajuna | Spain | 0.2093 | 0.7907 | 0.0420 | 0.0114 |
| SA01 | Sayaguesa | Spain | 0.2143 | 0.7857 | 0.0339 | *0.0021* |
| TU | Tudanca | Spain | 0.2773 | 0.7227 | 0.0382 | 0.1749 |
| TU01 | Tudanca | Spain | 0.2580 | 0.7420 | 0.0413 | 0.0295 |

*Appendix 2—table 1 Continued on next page*

*Appendix 2—table 1 Continued*

| Breed code | Breed | Origin | Proportion European aurochs | Proportion Anatolia Neolithic | Standard error | p-Value |
|---|---|---|---|---|---|---|
| WP01 | White Park | England | 0.2765 | 0.7235 | 0.0362 | 0.5794 |

Italic p-values indicate rejected models with p<0.05.

We notice that these results differ from the *D* statistic results in the literature *Upadhyay et al., 2017*, detected more aurochs alleles in breeds from the British Isles and Ireland, the Netherlands, Iberia, and Jersey when compared to Alpine breeds (Brown Swiss and Fleckvieh) based on *D* statistics grouping the breeds by geographical origin. We can replicate these results on the breed level with most tests showing significantly more allele sharing between aurochs and the Western European breeds relative to the Alpine breeds without any tests significantly pointing in the opposite direction (*Appendix 2—table 2*). Furthermore, *da Fonseca et al., 2019*, presented *D* statistics suggesting more allele sharing between aurochs and Angus, Holstein, and Jersey when compared to Iberian breeds. We can also largely reproduce this pattern by comparing Jersey and Holstein Friesian (Angus is not part of this study) with the Iberian breeds in our dataset (*Appendix 2—table 3*): out of 28 individual tests, two short strong (Z>3) support and seven show intermediate support (Z>2) for more aurochs alleles in the non-Iberian breeds while none of the tests is showing significant support of the opposite pattern.

**Appendix 2—table 2.** *D* statistics verifying the results of *Upadhyay et al., 2017*.
pop3 is a Western European breed while pop4 is an Alpine breed. Negative values indicate an excess of allele sharing between pop2 and pop3.

| pop1 | pop2 | pop3 | pop4 | D | SE | Z | n |
|---|---|---|---|---|---|---|---|
| YAK | CPC98 | EL01 | FL01 | –0.02652393 | 0.00456532 | –5.80987815 | 450246 |
| YAK | CPC98 | EL01 | BS01 | –0.03070768 | 0.00487875 | –6.29417402 | 450241 |
| YAK | CPC98 | GA01 | FL01 | –0.03044061 | 0.00392968 | –7.74633205 | 450234 |
| YAK | CPC98 | GA01 | BS01 | –0.03442162 | 0.00434763 | –7.91732385 | 450223 |
| YAK | CPC98 | WP01 | FL01 | –0.02563453 | 0.00397421 | –6.45021264 | 450035 |
| YAK | CPC98 | WP01 | BS01 | –0.02957367 | 0.00450710 | –6.56157650 | 450031 |
| YAK | CPC98 | HL01 | FL01 | –0.02842438 | 0.00387999 | –7.32588339 | 450320 |
| YAK | CPC98 | HL01 | BS01 | –0.03246308 | 0.00432781 | –7.50104292 | 450319 |
| YAK | CPC98 | KC01 | FL01 | –0.02672545 | 0.00390891 | –6.83706658 | 450352 |
| YAK | CPC98 | KC01 | BS01 | –0.03070730 | 0.00431715 | –7.11286432 | 450340 |
| YAK | CPC98 | JE01 | FL01 | –0.01497177 | 0.00407495 | –3.67409596 | 450360 |
| YAK | CPC98 | JE01 | BS01 | –0.01907032 | 0.00441154 | –4.32282854 | 450342 |
| YAK | CPC98 | DB01 | FL01 | –0.00874258 | 0.00384590 | –2.27322063 | 450195 |
| YAK | CPC98 | DB01 | BS01 | –0.01281078 | 0.00438547 | –2.92118480 | 450193 |
| YAK | CPC98 | DF01 | FL01 | –0.01004034 | 0.00368917 | –2.72157043 | 450373 |
| YAK | CPC98 | DF01 | BS01 | –0.01412771 | 0.00421435 | –3.35228684 | 450371 |
| YAK | CPC98 | HF01 | FL01 | –0.00512796 | 0.00341739 | –1.50055117 | 450378 |
| YAK | CPC98 | HF01 | BS01 | –0.00914451 | 0.00385993 | –2.36908732 | 450376 |
| YAK | CPC98 | MR01 | FL01 | –0.01566191 | 0.00346923 | –4.51451546 | 450313 |
| YAK | CPC98 | MR01 | BS01 | –0.01967925 | 0.00393753 | –4.99786229 | 450307 |
| YAK | CPC98 | AL01 | FL01 | –0.00515036 | 0.00420798 | –1.22395152 | 450178 |
| YAK | CPC98 | AL01 | BS01 | –0.00916224 | 0.00476407 | –1.92319693 | 450170 |
| YAK | CPC98 | AR01 | FL01 | –0.00733482 | 0.00358002 | –2.04881890 | 450360 |

*Appendix 2—table 2 Continued on next page*

*Appendix 2—table 2 Continued*

| pop1 | pop2 | pop3 | pop4 | D | SE | Z | n |
|------|------|------|------|---|----|----|----|
| YAK | CPC98 | AR01 | BS01 | −0.01140575 | 0.00397343 | −2.87050092 | 450357 |
| YAK | CPC98 | CC01 | FL01 | −0.01185512 | 0.00429875 | −2.75780703 | 450210 |
| YAK | CPC98 | CC01 | BS01 | −0.01592615 | 0.00467210 | −3.40877812 | 450196 |
| YAK | CPC98 | CL01 | FL01 | −0.00620692 | 0.00463521 | −1.33908070 | 449745 |
| YAK | CPC98 | CL01 | BS01 | −0.01051648 | 0.00500205 | −2.10243173 | 449745 |
| YAK | CPC98 | MI01 | FL01 | −0.00384595 | 0.00456479 | −0.84252575 | 450236 |
| YAK | CPC98 | MI01 | BS01 | −0.00801398 | 0.00496662 | −1.61356702 | 450221 |
| YAK | CPC98 | BC01 | FL01 | −0.00087993 | 0.00355382 | −0.24760203 | 450356 |
| YAK | CPC98 | BC01 | BS01 | −0.00498252 | 0.00395135 | −1.26096675 | 450345 |
| YAK | CPC98 | BN01 | FL01 | 0.00099552 | 0.00349974 | 0.28445469 | 450351 |
| YAK | CPC98 | BN01 | BS01 | −0.00310245 | 0.00405350 | −0.76537570 | 450348 |
| YAK | CPC98 | CA01 | FL01 | −0.00565510 | 0.00342051 | −1.65329057 | 450392 |
| YAK | CPC98 | CA01 | BS01 | −0.00977242 | 0.00389924 | −2.50623975 | 450386 |
| YAK | CPC98 | LI01 | FL01 | −0.00497786 | 0.00344966 | −1.44300022 | 450380 |
| YAK | CPC98 | LI01 | BS01 | −0.00909228 | 0.00392637 | −2.31569677 | 450374 |
| YAK | CPC98 | LM01 | FL01 | −0.01139534 | 0.00342571 | −3.32641246 | 450377 |
| YAK | CPC98 | LM01 | BS01 | −0.01553056 | 0.00399342 | −3.88903604 | 450373 |
| YAK | CPC98 | PA01 | FL01 | −0.00469758 | 0.00300994 | −1.56068935 | 450393 |
| YAK | CPC98 | PA01 | BS01 | −0.00883186 | 0.00363419 | −2.43021611 | 450384 |
| YAK | CPC98 | PA02 | FL01 | 0.00015340 | 0.00519442 | 0.02953081 | 449545 |
| YAK | CPC98 | PA02 | BS01 | −0.00393103 | 0.00545916 | −0.72008057 | 449548 |
| YAK | CPC98 | SA01 | FL01 | −0.00887584 | 0.00354305 | −2.50514043 | 450377 |
| YAK | CPC98 | SA01 | BS01 | −0.01299958 | 0.00401364 | −3.23885174 | 450369 |
| YAK | CPC98 | TU | FL01 | −0.00446867 | 0.00469083 | −0.95263975 | 449680 |
| YAK | CPC98 | TU | BS01 | −0.00849743 | 0.00509594 | −1.66748978 | 449674 |

**Appendix 2—table 3.** *D* statistics verifying the results of ***da Fonseca et al., 2019***.
pop3 is an Iberia breed while pop4 is a central or Northwestern European breed. Positive values indicate an excess of allele sharing between pop2 and pop4.

| pop1 | pop2 | pop3 | pop4 | D | SE | Z | n |
|------|------|------|------|---|----|----|----|
| YAK | CPC98 | AL01 | HF01 | 0.00010984 | 0.00444032 | 0.02473730 | 450186 |
| YAK | CPC98 | AL01 | JE01 | 0.00975475 | 0.00476772 | 2.04599760 | 450158 |
| YAK | CPC98 | AR01 | HF01 | −0.00209568 | 0.00376061 | −0.55727039 | 450365 |
| YAK | CPC98 | AR01 | JE01 | 0.00757226 | 0.00436317 | 1.73549446 | 450336 |
| YAK | CPC98 | CC01 | HF01 | −0.00664020 | 0.00434678 | −1.52761461 | 450211 |
| YAK | CPC98 | CC01 | JE01 | 0.00298956 | 0.00493465 | 0.60583115 | 450192 |
| YAK | CPC98 | CL01 | HF01 | −0.00089510 | 0.00478336 | −0.18712725 | 449748 |
| YAK | CPC98 | CL01 | JE01 | 0.00890081 | 0.00511564 | 1.73992026 | 449721 |
| YAK | CPC98 | MI01 | HF01 | 0.00119432 | 0.00474148 | 0.25188685 | 450238 |
| YAK | CPC98 | MI01 | JE01 | 0.01099329 | 0.00508326 | 2.16264559 | 450206 |
| YAK | CPC98 | BC01 | HF01 | 0.00420509 | 0.00380025 | 1.10653083 | 450359 |
| YAK | CPC98 | BC01 | JE01 | 0.01401724 | 0.00423143 | 3.31264957 | 450329 |

*Appendix 2—table 3 Continued on next page*

*Appendix 2—table 3 Continued*

| pop1 | pop2 | pop3 | pop4 | D | SE | Z | n |
|------|------|------|------|---|-----|---|---|
| YAK | CPC98 | BN01 | HF01 | 0.00610700 | 0.00367932 | 1.65981831 | 450360 |
| YAK | CPC98 | BN01 | JE01 | 0.01585867 | 0.00427758 | 3.70739043 | 450327 |
| YAK | CPC98 | CA01 | HF01 | –0.00046112 | 0.00364117 | –0.12663996 | 450396 |
| YAK | CPC98 | CA01 | JE01 | 0.00926087 | 0.00415092 | 2.23104269 | 450370 |
| YAK | CPC98 | LI01 | HF01 | 0.00017668 | 0.00374294 | 0.04720225 | 450385 |
| YAK | CPC98 | LI01 | JE01 | 0.00994126 | 0.00422452 | 2.35322801 | 450356 |
| YAK | CPC98 | LM01 | HF01 | –0.00611705 | 0.00379450 | –1.61208547 | 450383 |
| YAK | CPC98 | LM01 | JE01 | 0.00356800 | 0.00415838 | 0.85802696 | 450352 |
| YAK | CPC98 | PA01 | HF01 | 0.00044748 | 0.00328790 | 0.13609781 | 450395 |
| YAK | CPC98 | PA01 | JE01 | 0.01022468 | 0.00374297 | 2.73170629 | 450366 |
| YAK | CPC98 | PA02 | HF01 | 0.00519730 | 0.00529190 | 0.98212286 | 449549 |
| YAK | CPC98 | PA02 | JE01 | 0.01507871 | 0.00565298 | 2.66739024 | 449529 |
| YAK | CPC98 | SA01 | HF01 | –0.00363095 | 0.00362183 | –1.00251585 | 450382 |
| YAK | CPC98 | SA01 | JE01 | 0.00607429 | 0.00435903 | 1.39349602 | 450359 |
| YAK | CPC98 | TU | HF01 | 0.00055343 | 0.00493685 | 0.11210270 | 449674 |
| YAK | CPC98 | TU | JE01 | 0.01044897 | 0.00524529 | 1.99206629 | 449653 |

These *D* statistic results are an apparent contradiction to the qpAdm results. However, some of these tests contain breeds for which the two-source model was rejected by qpAdm (*Appendix 2—table 1*) including a breed used as 'reference' in the comparisons (Holstein Friesian). *D* statistics are known to be sensitive to certain biases (*Rojo et al., 2015*) including ghost admixture, i.e., gene flow from an unsampled population. Considering the complex history of commercial cattle breeds, it is possible that these specific breeds have received parts of their ancestry from another source than just ancient domestic European cattle and European aurochs. We tested this by running another set of qpAdm models moving *B. indicus* from the 'right' populations to the sources, i.e., testing each breed as a composition of Anatolian Neolithic cattle, European aurochs, and zebu cattle (*Appendix 2—table 4*). As a result, we now have fitting models for all breeds (p>0.01). The estimates of Zebu ancestry are all rather low, <12%. We assume that these models do not necessarily measure zebu ancestry in all breeds but more generally detect non-European ancestry in commercial breeds. Estimates of aurochs ancestry vary much more than in the two-source models which can be partly explained by the increased uncertainties (SEs up to 5%) which also has the consequence that all breeds still have overlapping 95% confidence intervals.

**Appendix 2—table 4.** qpAdm results for the modern breeds in a three-source model.

| Breed | European Aurochs proportion | European domestic proportion | Indicus proportion | SE (aurochs) | SE (*taurus*) | SE (*indicus*) | p |
|-------|------|------|------|------|------|------|------|
| AL01 | 0.1489 | 0.8332 | 0.0179 | 0.0477 | 0.0416 | 0.0086 | 0.1290 |
| AN01 | 0.1334 | 0.8403 | 0.0264 | 0.0505 | 0.0439 | 0.0096 | 0.6038 |
| AR01 | 0.1465 | 0.8333 | 0.0202 | 0.0427 | 0.0370 | 0.0081 | 0.2356 |
| BC01 | 0.1446 | 0.8303 | 0.0252 | 0.0433 | 0.0378 | 0.0080 | 0.5834 |
| BK01 | 0.0399 | 0.8771 | 0.0829 | 0.0489 | 0.0426 | 0.0093 | 0.0774 |
| BN01 | 0.1223 | 0.8447 | 0.0329 | 0.0430 | 0.0375 | 0.0080 | 0.1930 |
| BS01 | 0.0690 | 0.9010 | 0.0300 | 0.0477 | 0.0416 | 0.0088 | 0.1299 |
| BU01 | 0.0683 | 0.8212 | 0.1106 | 0.0465 | 0.0405 | 0.0088 | 0.1236 |
| BU02 | 0.0481 | 0.8906 | 0.0613 | 0.0464 | 0.0403 | 0.0086 | 0.1336 |
| CA01 | 0.1713 | 0.8153 | 0.0135 | 0.0426 | 0.0370 | 0.0079 | 0.1055 |

*Appendix 2—table 4 Continued on next page*

*Appendix 2—table 4 Continued*

| Breed | European Aurochs proportion | European domestic proportion | Indicus proportion | SE (aurochs) | SE (*taurus*) | SE (*indicus*) | p |
|---|---|---|---|---|---|---|---|
| CC01 | 0.1525 | 0.8241 | 0.0233 | 0.0492 | 0.0425 | 0.0092 | 0.5879 |
| CH01 | 0.0313 | 0.8667 | 0.1020 | 0.0492 | 0.0429 | 0.0092 | 0.1833 |
| CL01 | 0.1003 | 0.8756 | 0.0241 | 0.0518 | 0.0452 | 0.0094 | 0.2378 |
| DB01 | 0.1892 | 0.7951 | 0.0158 | 0.0449 | 0.0390 | 0.0084 | 0.4417 |
| DF01 | 0.1837 | 0.7962 | 0.0201 | 0.0444 | 0.0386 | 0.0084 | 0.4395 |
| FL01 | 0.1138 | 0.8590 | 0.0271 | 0.0427 | 0.0372 | 0.0080 | 0.1621 |
| GA01 | 0.2064 | 0.7859 | 0.0077 | 0.0432 | 0.0377 | 0.0080 | 0.6631 |
| GW01 | 0.1925 | 0.7969 | 0.0107 | 0.0442 | 0.0386 | 0.0081 | 0.3708 |
| HF01 | 0.1941 | 0.7799 | 0.0260 | 0.0430 | 0.0374 | 0.0080 | 0.0834 |
| HL01 | 0.1905 | 0.8023 | 0.0072 | 0.0459 | 0.0400 | 0.0087 | 0.0260 |
| JE01 | 0.1233 | 0.8527 | 0.0240 | 0.0458 | 0.0399 | 0.0086 | 0.3298 |
| KC01 | 0.2136 | 0.7775 | 0.0089 | 0.0438 | 0.0382 | 0.0081 | 0.5727 |
| LI01 | 0.1345 | 0.8454 | 0.0201 | 0.0433 | 0.0377 | 0.0081 | 0.6682 |
| LM01 | 0.1640 | 0.8229 | 0.0131 | 0.0429 | 0.0374 | 0.0081 | 0.5898 |
| MA01 | 0.0401 | 0.8589 | 0.1010 | 0.0441 | 0.0384 | 0.0087 | 0.1193 |
| ME01 | 0.1554 | 0.8229 | 0.0217 | 0.0432 | 0.0377 | 0.0081 | 0.3929 |
| MI01 | 0.1468 | 0.8312 | 0.0220 | 0.0489 | 0.0426 | 0.0090 | 0.6720 |
| MN01 | 0.1336 | 0.8423 | 0.0240 | 0.0458 | 0.0399 | 0.0086 | 0.1144 |
| MR01 | 0.1525 | 0.8276 | 0.0198 | 0.0443 | 0.0387 | 0.0082 | 0.3582 |
| PA01 | 0.1437 | 0.8364 | 0.0199 | 0.0412 | 0.0357 | 0.0078 | 0.1722 |
| PA02 | 0.1864 | 0.8000 | 0.0135 | 0.0513 | 0.0444 | 0.0098 | 0.6806 |
| PO01 | 0.0003 | 0.8877 | 0.1119 | 0.0557 | 0.0485 | 0.0112 | 0.6150 |
| RO01 | 0.0318 | 0.8844 | 0.0838 | 0.0461 | 0.0402 | 0.0090 | 0.3009 |
| SA01 | 0.1260 | 0.8532 | 0.0208 | 0.0430 | 0.0375 | 0.0079 | 0.1805 |
| TU | 0.1698 | 0.8105 | 0.0196 | 0.0500 | 0.0434 | 0.0094 | 0.3151 |
| TU01 | 0.1303 | 0.8449 | 0.0248 | 0.0532 | 0.0465 | 0.0096 | 0.2313 |
| WP01 | 0.1637 | 0.8184 | 0.0179 | 0.0461 | 0.0402 | 0.0085 | 0.5317 |

Consequently, we believe that the D statistic results are at least partly driven by low levels of non-European ancestry in many commercial breeds. qpAdm, in contrast, did not just allow us to obtain quantitative estimates of aurochs ancestry but also to identify cases where the two sources did not fit the data. The confidence intervals for each breed are still rather wide, highlighting the need for more high-quality reference data from European aurochs as well as from informative groups that can be used as "right" populations for such analyses.

## Appendix 3

### Stable isotope data

Previous studies comparing the stable isotope values of domestic cattle and aurochs have noted differences between the species. A study of material excavated from the UK (*Lynch et al., 2008*) noted an overall difference of 1‰ in the carbon isotope values, with aurochs being the most depleted in carbon-13. There was overlap in the values for the two species, but at sites where both species were found this difference ranged from 0.3‰ to 1.8‰. The overlap in the nitrogen isotope values was large and difficult to interpret. *Lynch et al., 2008*, interpret their data as reflecting different niches for the two species with domestic cattle in more open settings, while the aurochs could be in more forested areas (the canopy effect on carbon stable isotopes), or in wet ground (that might have a similar effect of depleting carbon isotopes). Consequently, *Lynch et al., 2008*, were suggesting that stable isotope data could be used to infer niche separation between the species, based on human management of the domestic cattle, separating them from their wild counterparts. For material excavated from sites in Denmark and Northern Germany (*Noe-Nygaard et al., 2005*), the situation is less clear due to the overlap in the values. However, aurochs values change over time (–20‰ to –24‰), which might reflect environmental change. The distinctions between the domestic cattle and aurochs in the above studies were based on morphological and size differences.

By comparison, we have used the material here to explore niche separation and/or other differences between wild and domestic cattle in the Iberian context. The data produced from the stable isotope analysis is given in *Supplementary file 1* – stable isotopes, with additional previously published data. We have compared our data to that from published studies, spanning the Mesolithic to the Bronze Age (*Fernández-Crespo and Schulting, 2017*; *Fontanals-Coll et al., 2015*; *Garcia Guixé et al., 2006*; *Jones et al., 2019*; *Navarrete et al., 2017*; *Salazar-García, 2009*; *Salazar-García, 2011*; *Salazar-García et al., 2014*; *Sarasketa-Gartzia et al., 2018*; *Villalba-Mouco et al., 2018*; *Fernández-Crespo et al., 2019*). While the stable isotope data is directly comparable, it should be noted that these different studies have used different skeletal elements from each other (sometimes horn, tooth, or different types of bone) depending on availability, and have relied on morphology/size to determine species and sometimes species was only determined as *Bos sp*. Thus, there will be some inconsistency between studies as to how species has been determined, ostensibly they may have used different criteria or subjective criteria. Dates for these samples were either given in the publications or have been assigned here based on dates from related/associated material (often human samples dated from the same sites). Approximate latitudes and longitudes have also been assigned to the sites. Samples with C:N ratios of 3.7 were excluded from further analysis and in addition, to constrain analysis to likely C3 ecosystems only, three *B. taurus* samples from southern Spain (S-EVA 9042, S-EVA 7377, S-EVA 7378) have been excluded as they likely consumed some $C_4$ plants in their diet (*Salazar-García, 2009*; *Salazar-García, 2011*). For this analysis, the samples from El Callado described in the publication as *Bos species Salazar-García, 2011* have been considered as *B. primigenius*, as they date to the Mesolithic.

The samples in our analysis were also from varied skeletal elements and originally classified on morphology to *B. taurus*, *B. primigenius,* or not classified. We can consider these as similar to those in the published literature, with 'not classified' equivalent to *Bos sp.*

### Analysis based on morphology

Comparing $^{15}$N against $^{13}$C, the data does not reveal any obvious distinction between the groups (see *Appendix 3—figure 1*), other than the *B. primigenus* tend to have higher $^{13}$C values and lower $^{15}$N values than *B. taurus* (only one *B. primigenius* sample has an $^{15}$N value above 6.4‰). Testing the normality of the distributions was carried out in RStudio using Shapiro-Wilk and Anderson-Darling tests (using nortest package) (see *Appendix 3—table 1*). In most cases, at least one dataset to be

used in statistical comparisons was considered non-normally distributed, so comparisons have been carried out using a Wilcoxon rank sum test (Mann-Whitney test). All statistical analysis here was performed in RStudio, Version 2024.09.0+375.

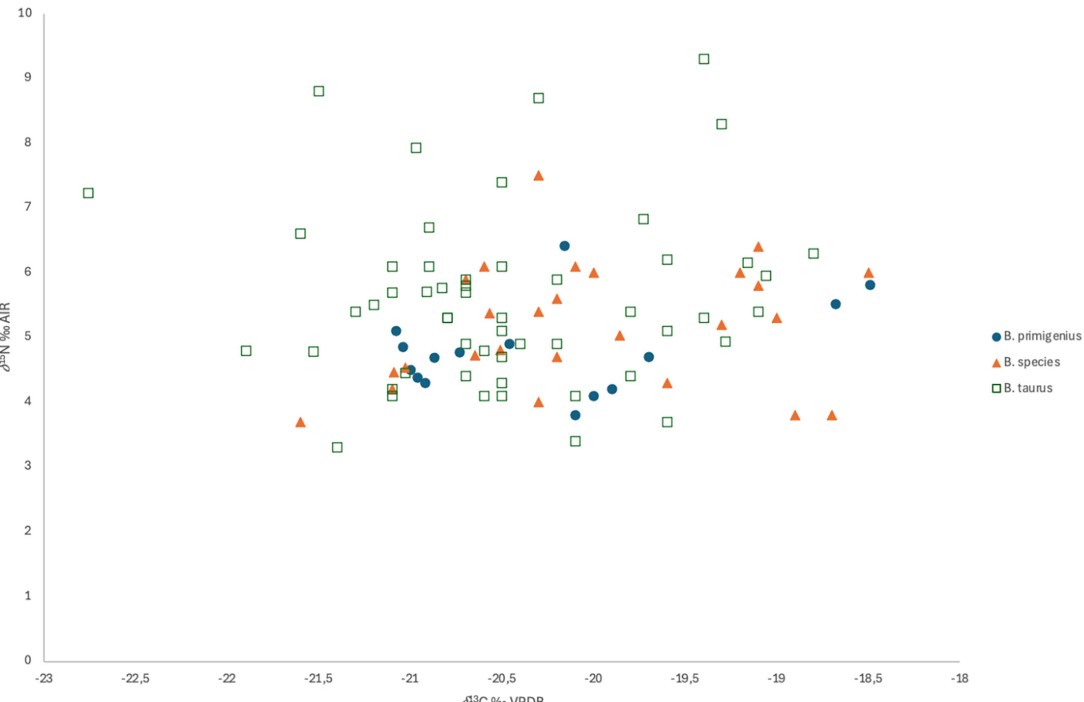

**Appendix 3—figure 1.** The distribution of $^{15}$N vs $^{13}$C for the samples, identified on the basis of morphology.

Wilcoxon rank sum tests of various groups reveal no significant differences in the carbon isotope values between the groups tested *B. primigenius* vs *B. taurus*, of when *B. primigenius* is included with likely *B. primigenius* and or tested against *B. taurus* and *B. species* as one group (see *Appendix 3— table 2* for W- and p-values). Statistically significant differences are observed when comparing nitrogen isotope values between (*B. primigenius* vs *B. taurus*, *B. primigenius* vs *B. taurus* and *B. species* and when 'likely *B. primigenius*' is added to the *B. primigenius* group in both comparisons). It should be noted that the stable isotope values are influenced by the environment and we can anticipate geographical differences in their distribution. Of importance here is that the geographic distribution of the species is not even across latitude and it can thus influence our observations. Six of the 13 aurochs (46%) are below 41°N, while only 3 of 54 (approx. 6%) of *B. taurus* are below this latitude. The $^{13}$C values are higher at lower latitudes and the distribution of carbon isotope values is narrower at lower latitudes as well, with no values lower than –20.1‰ observed south of 41°N. It should be noted most sites are north of 41°N and thus these differences could be a sampling artefact (especially the narrow range of values in the south), but as noted that the southern sites have some of the highest carbon isotope values and this would be expected given climatic/ecological differences between the north and south of Spain. Nevertheless, in the northern sites only (>41°N), where most samples have been recovered the mean carbon stable isotope values of *B. primigenus* and *B. taurus* are not statistically different.

For nitrogen isotopes, there is no linear relationship between latitude and $^{15}$N, and the range of values is similar across latitudes, except that values of 6.7‰ or greater are only observed north of 42° latitude. When comparing geographically only those animals recovered from >42°N there are no statistical differences between groups for $^{15}$N. The average $^{15}$N value for all samples is 5.35‰ (sd 1.18), there are 11 *B. taurus* and *B. sp.* samples that fall outside the average +1 sd and no aurochs. This might reflect some level of management in the domestic fauna (corralling or feeding of manured crops or pastures).

**Appendix 3—table 1.** Collagen isotope data summary based on morphological identification. Significant differences (at p=0.05) from normal distributions are highlighted in blue.

| | Morphological species | Number of samples | Mean | Std dev | Median | Shapiro-Wilk W-value | Shapiro-Wilk p-value | Anderson-Darling A-value | Anderson-Darling p-value |
|---|---|---|---|---|---|---|---|---|---|
| | | | $^{13}C$ | | | Normality tests | | | |
| | B. taurus | 54 | −20.48 | 0.80 | −20.55 | 0.972 | 0.242 | 0.628 | 0.097 |
| | B. species | 26 | −20.02 | 0.83 | −20.20 | 0.961 | 0.415 | 0.438 | 0.273 |
| | B. primigenius | 13 | −20.20 | 0.86 | −20.16 | 0.864 | 0.043 | 0.670 | 0.061 |
| | B. primigenius+B. primigenius? | 15 | −20.27 | 0.83 | −20.46 | 0.853 | 0.019 | 0.784 | 0.032 |
| All | B. taurus+B. species | 80 | −20.33 | 0.83 | −20.50 | 0.976 | 0.133 | 0.810 | 0.035 |
| | B. taurus | 51 | −20.56 | 0.75 | −20.60 | 0.977 | 0.403 | 0.529 | 0.168 |
| | B. species | 21 | −20.26 | 0.73 | −20.30 | 0.955 | 0.415 | 0.429 | 0.282 |
| | B. primigenius | 7 | −20.82 | 0.31 | −20.92 | 0.775 | 0.023 | N/A | N/A |
| | B. primigenius+B. primigenius? | 9 | −20.80 | 0.31 | −20.92 | 0.835 | 0.050 | 0.662 | 0.055 |
| Northern (>41°N) | B. taurus+B. species | 72 | −20.47 | 0.75 | −20.51 | 0.979 | 0.259 | 0.589 | 0.120 |
| | | | $^{15}N$ | | | Normality tests | | | |
| | B. taurus | 54 | 5.58 | 1.33 | 5.40 | 0.940 | 0.010 | 0.963 | 0.014 |
| | B. species | 26 | 5.18 | 0.95 | 5.25 | 0.961 | 0.420 | 0.317 | 0.519 |
| | B. primigenius | 13 | 4.79 | 0.74 | 4.69 | 0.927 | 0.310 | 0.430 | 0.261 |
| | B. primigenius+B. primigenius? | 15 | 4.80 | 0.69 | 4.70 | 0.938 | 0.361 | 0.397 | 0.324 |
| All | B. taurus+B. species | 80 | 5.45 | 1.23 | 5.34 | 0.943 | 0.001 | 1.096 | 0.007 |
| | B. taurus | 46 | 5.60 | 1.41 | 5.40 | 0.935 | 0.013 | 0.975 | 0.013 |
| | B. species | 11 | 5.36 | 0.92 | 5.03 | 0.862 | 0.061 | 0.575 | 0.104 |
| | B. primigenius | 7 | 4.88 | 0.73 | 4.69 | 0.778 | 0.024 | N/A | N/A |
| | B. primigenius+B. primigenius? | 9 | 4.88 | 0.63 | 4.77 | 0.776 | 0.011 | 0.821 | 0.020 |
| Northern (>42°N) | B. taurus+B. species | 57 | 5.55 | 1.33 | 5.38 | 0.930 | 0.003 | 1.291 | 0.002 |

**Appendix 3—table 2.** Summary of Wilcoxon rank sum test results of comparisons of stable isotope data based on morphological characterisation.
Significant differences (at p=0.05) are highlighted in blue.

| | Test comparison | | W-value | p-value |
|---|---|---|---|---|
| | | | Wilcoxon rank sum test with continuity correction | |
| | B. primigenius vs B. taurus | $^{13}C$ | 398 | 0.461 |
| | | $^{15}N$ | 213 | 0.029 |
| | B. primigenius vs B. taurus+ B. species | $^{13}C$ | 549 | 0.752 |
| | | $^{15}N$ | 339 | 0.045 |
| | B. primigenius+ B. primigenius? vs B. taurus | $^{13}C$ | 444 | 0.575 |
| | | $^{15}N$ | 246 | 0.021 |
| | B. primigenius+B. primigenius? vs B. taurus +B. species | $^{13}C$ | 607 | 0.947 |
| All | | $^{15}N$ | 395 | 0.036 |

*Appendix 3—table 2 Continued on next page*

*Appendix 3—table 2 Continued*

| | | | Wilcoxon rank sum test with continuity correction | |
|---|---|---|---|---|
| | *B. primigenius* vs *B. taurus* | $^{13}C$ | 128 | 0.232 |
| | | $^{15}N$ | 105 | 0.141 |
| | *B. primigenius* vs *B. taurus+B. species* | $^{13}C$ | 165 | 0.135 |
| | | $^{15}N$ | 128 | 0.124 |
| | *B. primigenius+B. primigenius?* vs *B. taurus* | $^{13}C$ | 174 | 0.254 |
| | | $^{15}N$ | 136 | 0.106 |
| | *B. primigenius+B. primigenius?* vs *B. taurus+B. species* | $^{13}C$ | 223 | 0.131 |
| Northern only | | $^{15}N$ | 169 | 0.102 |

## Analysis based on genetics

Considering the genetic analysis of our samples, some samples can be reclassified when compared with the morphological categories. Using the genetic data we have established three models for the categorisation of the animals for stable isotope data analysis. Model 1: categorises the animal on its majority genetic ancestry, so if an animal is greater than 50% aurochs, it is considered aurochs, otherwise domestic. Model 2: categorises the animal if it is in the top 30th percentile of its ancestry. So that 70% or above of aurochs ancestry = aurochs; 30% or below is considered taurine and anything in between is a hybrid. Model 3: is the same as 2 with a 20% cutoff. All other published data have been included in the analysis, retaining their morphological assignments to species. Our samples with too little DNA for analysis have been excluded. Making the same comparisons of the stable isotope data as above using the new categories, we can explore if the reassignment has made a difference (summarised in *Appendix 3—table 3*). Similar to the morphological distinctions, no datasets pass both tests for data normality, so subsequent inferential analysis was made using Mann-Whitney U tests (see *Appendix 3—table 4*). Comparing *B. primigenius* vs *B. taurus*, defined using the models above, there are no statistical differences for any model considering all the data, nor only the northern datasets.

**Appendix 3—table 3.** Collagen isotope data summary based on genetic (or morphological identification for published studies).

Significant differences (at p=0.05) from normal distributions are highlighted in blue.

| | Genetic model | Genetic species | Number of samples | $^{13}C$ Mean | Std dev | Median | Normality tests Shapiro-Wilk W-value | Shapiro-Wilk p-value | Anderson-Darling A-value | Anderson-Darling p-value |
|---|---|---|---|---|---|---|---|---|---|---|
| | | *B. taurus* | 55 | −20.48 | 0.73 | −20.65 | 0.951 | 0.025 | 1.140 | 0.005 |
| | Model 1 | *B. primigenius* | 17 | −20.38 | 0.99 | −20.46 | 0.937 | 0.283 | 0.470 | 0.216 |
| | | *B. taurus* | 52 | −20.45 | 0.74 | −20.60 | 0.958 | 0.064 | 0.916 | 0.018 |
| | | *B. primigenius* | 16 | −20.23 | 0.81 | −20.31 | 0.884 | 0.045 | 0.612 | 0.092 |
| | Model 2 | Hybrid | 4 | −21.41 | 0.90 | −21.03 | 0.719 | 0.019 | N/A | N/A |
| | | *B. taurus* | 51 | −20.44 | 0.75 | −20.60 | 0.960 | 0.084 | 0.849 | 0.027 |
| | | *B. primigenius* | 11 | −20.08 | 0.89 | −20.10 | 0.893 | 0.150 | 0.478 | 0.187 |
| All | Model 3 | Hybrid | 10 | −20.93 | 0.76 | −20.94 | 0.826 | 0.030 | 0.867 | 0.016 |

*Appendix 3—table 3 Continued on next page*

*Appendix 3—table 3 Continued*

| | | | | **¹³C** | | | | **Normality tests** | | |
|---|---|---|---|---|---|---|---|---|---|---|
| | Model 1 | *B. taurus* | 52 | –20.55 | 0.68 | –20.70 | 0.959 | 0.069 | 0.911 | 0.019 |
| | | *B. primigenius* | 11 | –20.87 | 0.76 | –20.92 | 0.852 | 0.045 | 0.74 | 0.038 |
| | | *B. taurus* | 49 | –20.53 | 0.69 | –20.65 | 0.966 | 0.164 | 0.723 | 0.055 |
| | | *B. primigenius* | 10 | –20.68 | 0.45 | –20.90 | 0.851 | 0.060 | 0.636 | 0.068 |
| | Model 2 | Hybrid | 4 | –21.41 | 0.90 | –21.03 | 0.719 | 0.019 | N/A | N/A |
| | | *B. taurus* | 48 | –20.52 | 0.69 | –20.63 | 0.968 | 0.209 | 0.668 | 0.076 |
| | | *B. primigenius* | 5 | –20.81 | 0.37 | –20.92 | 0.763 | 0.039 | N/A | N/A |
| Northern (>41°N) | Model 3 | Hybrid | 10 | –20.93 | 0.76 | –20.94 | 0.826 | 0.030 | 0.867 | 0.016 |

| | Genetic model | Genetic species | Number of samples | **¹⁵N** Mean | Std dev | Median | **Normality tests** Shapiro-Wilk W-value | Shapiro-Wilk p-value | Anderson-Darling A-value | Anderson-Darling p-value |
|---|---|---|---|---|---|---|---|---|---|---|
| | Model 1 | *B. taurus* | 55 | 5.48 | 1.31 | 5.30 | 0.915 | 0.001 | 1.424 | 0.001 |
| | | *B. primigenius* | 17 | 5.06 | 0.99 | 4.70 | 0.898 | 0.062 | 0.702 | 0.054 |
| | | *B. taurus* | 52 | 5.51 | 1.33 | 5.30 | 0.918 | 0.002 | 1.325 | 0.002 |
| | | *B. primigenius* | 16 | 4.93 | 0.85 | 4.70 | 0.914 | 0.133 | 0.557 | 0.126 |
| | Model 2 | Hybrid | 4 | 5.50 | 1.31 | 5.15 | 0.877 | 0.326 | N/A | N/A |
| | | *B. taurus* | 51 | 5.51 | 1.34 | 5.30 | 0.918 | 0.002 | 1.324 | 0.002 |
| | | *B. primigenius* | 11 | 4.77 | 0.81 | 4.50 | 0.893 | 0.151 | 0.572 | 0.106 |
| All | Model 3 | Hybrid | 10 | 5.40 | 0.99 | 5.10 | 0.885 | 0.148 | 0.480 | 0.179 |
| | Model 1 | *B. taurus* | 47 | 5.47 | 1.38 | 5.30 | 0.903 | 0.001 | 1.518 | 0.001 |
| | | *B. primigenius* | 11 | 5.26 | 1.05 | 4.90 | 0.816 | 0.015 | 0.897 | 0.014 |
| | | *B. taurus* | 44 | 5.51 | 1.41 | 5.30 | 0.908 | 0.002 | 1.381 | 0.001 |
| | | *B. primigenius* | 10 | 5.07 | 0.87 | 4.80 | 0.799 | 0.014 | 0.875 | 0.015 |
| | Model 2 | Hybrid | 4 | 5.50 | 1.31 | 5.15 | 0.877 | 0.326 | N/A | N/A |
| | | *B. taurus* | 43 | 5.51 | 1.43 | 5.30 | 0.907 | 0.002 | 1.379 | 0.001 |
| | | *B. primigenius* | 5 | 4.87 | 0.88 | 4.50 | 0.700 | 0.010 | N/A | N/A |
| Northern (>42°N) | Model 3 | Hybrid | 10 | 5.40 | 0.99 | 5.10 | 0.885 | 0.148 | 0.480 | 0.179 |

**Appendix 3—table 4.** Summary of Wilcoxon rank sum test results of comparisons of stable isotope data based on morphological characterisation (or morphological identification for published studies).

| | Test comparison | | | **Wilcoxon rank sum test with continuity correction** W-value | p-Value |
|---|---|---|---|---|---|
| | | Model 1 | ¹³C | 516 | 0.524 |
| | | | ¹⁵N | 370 | 0.198 |
| | | Model 2 | ¹³C | 478 | 0.374 |
| | | | ¹⁵N | 298 | 0.089 |
| | | Model 3 | ¹³C | 342 | 0.261 |
| All | *B. primigenius* vs *B. taurus* | | ¹⁵N | 177 | 0.056 |
| | | Model 1 | ¹³C | 237 | 0.379 |
| | | | ¹⁵N | 235 | 0.641 |
| | | Model 2 | ¹³C | 217 | 0.578 |
| | | | ¹⁵N | 178 | 0.349 |
| | | Model 3 | ¹³C | 87 | 0.322 |
| Northern only | *B. primigenius* vs *B. taurus* | | ¹⁵N | 74 | 0.258 |

## Conclusions

The values of the descriptive statistics of the stable isotope data here are different depending on how our samples have been categorised and grouped with the published data, whether by morphology or genetics. However, using non-parametric tests, a significant difference can only be observed when comparing between the $^{15}$N values of *B. primigenius* vs *B. taurus* (and *B. primigenius* vs *B. taurus*, *B. primigenius* vs *B. taurus* and *Bos species* and when 'likely *B. primigenius*' is added to the *B. primigenius* group in both comparisons) defined by morphology. When considering only sites north of 42°N this difference is no longer significant. All other comparisons, whether using a morphological or a genetic characterisation of the individuals, are not significant.

The most obvious discriminating feature of the data is that of the distribution of $^{15}$N values for taurine cattle and aurochs. This overlaps considerably, however, some taurine samples have $^{15}$N values greater than 6.5‰, while aurochs samples do not. This can be interpreted as both types of cattle often share an ecological niche, but that some taurine cattle had habitual access to the areas or resources not available to the *B. primigenius*. This could indicate that in some instances the domestic cattle are managed by humans in certain ways (corralled on manured ground, fed manured crops?), but often not. It should be noted that there is wide variation in the $^{15}$N within sites, so if this interpretation is true, not all cattle at a particular site are treated in the same manner. If we consider access to high nitrogen resources as a human management process, we can imply that aurochs are excluded from this, but that many cattle are in similar ecosystems as the aurochsen.

The observation here that the interpretation changes depending on the categorisation of the samples represents a challenge to scientists using this kind of data. The genetic data can allow us to more easily observe what we can describe as aurochs-taurine hybrids and even quantify a degree of hybridisation, but ancient management strategies may have been oblivious to the cattle ancestry, more concerned with the phenotypic attributes of the cattle (size temperament and manageability), in line with the morphological/metric and contextual assignment of species made in the present. Ancient farmers would have no knowledge of genetics, perhaps some knowledge of bloodlines, but would potentially herd/manage cattle that were useful and manageable, the latter might largely depend on size and temperament.

Contextualising our data with the UK and Danish/German datasets, there are some notable differences. In Northern Europe, average $^{15}$N is lower in *B. taurus* relative to *B. primigenius* and the opposite pattern to that observed in Iberia. While the UK and Danish/German *B. taurus* have similar absolute $^{15}$N values to those in Iberia. The Northern European individuals have lower $^{13}$C values than their Iberian counterparts (attributable to climatic and ecological differences), and the *B. primigenius* in Northern Europe have the lower $^{13}$C values compared to *B. taurus*. In Iberia this distinction between species is unclear. Considering only the northern Iberian sites *B. primigenius* is on average lower than *B. taurus* for $^{13}$C, but considering all sites this is reversed. This may reflect similarities in the environment of northern Iberia and the rest of Northern Europe (wetter and cooler) when compared to the south of Iberia. The significance of these differences is not clear, but considering the data as gross categories, it does appear that aurochs were confined to high nitrogen isotope areas in Northern Europe, but low $^{15}$N areas in Spain, giving clear evidence for the need to interpret and contextualise such data regionally (this would need some reassessment, if in the future we find, evidence for long-distance movement of either species).

