## [Editor Report · eLife Assessment]

Using genomic data from ancient and modern samples, this **important** study investigates the genomic history of cattle in Iberia, focusing on the admixture between domestic cattle and their wild ancestors, aurochs. The authors present **convincing** evidence for interbreeding between domestic cattle and wild aurochs since the Neolithic period, although the evidence of sex-biased introgression is weak. The authors also show that the aurochs ancestry in cattle stabilized at ~20% since ~4000 years ago and continues into modern breeds; however, the aurochs ancestry is not heightened in a modern breed of Spanish fighting bulls that are bred for aggressiveness. The work will be of interest to evolutionary biologists and quantitative geneticists who seek to understand the genomic history and genetic basis of trait variation of domesticated animals.

---

## [Referee Report · Reviewer #2 (Public review)]

Summary:

In this paper, the authors investigated the admixture history of domestic cattle since they were introduced into Iberia, by studying genomic data from 24 ancient samples dated to ~2000-8000 years ago and comparing them to modern breeds. They aimed to (1) characterize genomic variation of skeletal remains and concordance (or discordance) with morphological features; (2) test for hybridization between wild aurochs and domestic cattle; (3) test for correlation between genetic ancestry and stable isotope levels (which are indicative of ecological niche); and (4) test for previously hypothesized higher aurochs ancestry in a modern breed of fighting bulls.

Strengths:

Overall, this study collects valuable new data and tests several important hypotheses regarding the evolutionary history and genomic variation of domestic cattle in Iberia, such as admixture between domestic and wild populations, and correlation between genome-wide aurochs ancestry and aggressiveness.

Weaknesses:

Most conclusions are well supported by the data presented, with the strengths and caveats of each analysis clearly explained. The presence of admixed individuals in prehistorical periods strongly support hybridization between wild and domestic populations, although the evidence for sex-biased introgression and ecological niche sharing is relatively weak. Lastly, the authors presented convincing evidence for relatively constant aurochs ancestry across all modern breeds, including the Lidia breed that has been bred for aggressiveness for centuries.

Major comments:

As the authors pointed out, a major limitation of this study is uncertainty in the "population identity" for most sampled individuals (i.e., whether an individual belonged to the domesticated or wild herd when they were alive). Based on chronology, morphology and genetic data, it is clear the Mesolithic samples from the Artusia and Mendandia sites are bona fide aurochs, but the "population identities" of individuals from the other two sites are much less certain. Indeed, archeological and morphological evidence from El Portalon supports the presence of both domestic animals and wild aurochs, which is echoed by the inter-individual heterogeneity in genetic ancestry. Despite the strong evidence of hybridization, it is unclear whether these admixed individuals were raised in the domestic population or lived in the wild population and hunted, limiting the authors' ability to draw conclusions regarding the direction of gene flow.

In general, detecting sex-bias admixture is an inherently challenging problem, especially given limited data. The differential ancestry proportions (estimated by f4 ratios) on autosomes and X chromosome are indicative of sex-biased hybridization and consistent with previous mtDNA results and other non-genetic data. However, as shown in Fig 3, the confidence intervals of X and autosomal estimates overlap for all but a couple of individuals, despite the overall trend of the point estimates. Moreover, even if there is significant difference, it only suggests existence of sex-bias but does not speak to the extent (unless further quantitative argument is made). Statements such as "it was mostly aurochs males who contributed wild ancestry to domestic herds" is too strong and may be interpreted as extreme bias. The authors did a good job noting the caveats of this analysis and down-toned the statement in the main text, but claims regarding sex-bias hybridization that use the phrase "mostly" in the abstract and discussion need to be further weakened.

The stable isotope analysis is very under-powered, due to issues of categorization of wild vs domestic Bos, as discussed by the authors. Although the considerable overlap in stable isotope values between domestic and wild groups is consistent with shared ecological niche, but the absence of evidence (ie significant difference between groups) is not evidence of absence. Two alternative, non-mutually exclusive scenarios are (1) prevalent errors in classification of wild vs domestic individuals; (2) different ecological niches share similar isotope profiles. Thus, the claim "suggesting that wild and domesticated groups often did not occupy different niches in Iberia" is still too strong.

---

## [Referee Report · Reviewer #3 (Public review)]

Summary:

Günther and colleagues leverage ancient DNA data to track the genomic history of one of the most important farm animals (cattle) in Iberia, a region showing peculiarities both in terms of cultural practices as well as a climatic refugium during the LGM, the latter of which could have allowed the survival of endemic lineages. They document interesting trends of hybridisation with wild aurochs over the last 8-9 millennia, including a stabilisation of auroch ancestry ~4000 years ago, at ~20%, a time coincidental with the arrival of domestic horses from the Pontic steppe. Modern breeds such as the iconic Lidia used in bullfighting or bull running retain a comparable level of auroch ancestry.

Strengths:

The generation of ancient DNA data has been proven crucial to unravel the domestication history of traditional livestock, and this is challenging due to the environmental conditions of the Iberian peninsula, less favourable to DNA preservation. The authors leverage samples unearthed from key archaeological sites in Spain, including the karstic system of Atapuerca. Their results provide fresher insights into past management practices and permit characterisation of significant shifts in hybridization with wild aurochs.

Comments on revisions:

The authors have satisfactorily addressed my previous concerns. Last questions:

- How many MCMC iterations were run for Structf4? Can they show the likelihood of the last 10% of MCMC iterations? The results seem way too different for K = 4 vs. K = 5, but only for moo014 and moo019.

- I guess the authors also lack an "a" superindex in Table 1 for moo019.

- That Gyu2-related ancestry appears systematically for K=5 suggests that the Caucasus-related ancestry was already present in the pool that led to domesticates. Is it not important to discuss the implications of this possibility, for future analyses?

- If monophyletic, why choose between Bed3 and CPC98 if both could be combined as a single population to further reduce qpAdm and f4 confidence intervals?

- Why not combine all auroch Iberian samples as a single population for testing gene flow from this whole group of samples to ancient Iberian cattle? Would be the resulting coverage still too low?

- What is subindex 1 in the denominator of the f4 ratio (main methods)?

Thanks for your efforts

---

## [Author Response]

The following is the authors’ response to the original reviews.

**Public Reviews:**

**Reviewer #1 (Public Review):**
My main concern is the use of the 700K SNP dataset. This set of SNPs suffers from a heavy ascertainment bias, which can be seen in the PCA in the supplementary material where all the aurochs cluster in the center within the variation of cattle. Given the coverage of some of the samples, multiple individuals would have less than 10K SNP covered. The majority of these are unlikely to be informative here given that they would just represent fixed positions between taurine and indicine or SNPs mostly variable in milk cattle breeds. The authors would get a much better resolution (i.e. many more SNPs to work with their very low genome coverage data) using the 1000 bull genome project VCF data set:https://www.ebi.ac.uk/ena/browser/view/PRJEB42783 which based on whole genome resequencing data from many cattle. This will certainly help with improving the resolution of qpAdm and f4 analysis, which have huge confidence intervals in most cases. Right now some individuals have huge confidence intervals ranging from 0 to 80% auroch ancestry...

We thank the reviewer for this suggestion. We repeated our analyses with a SNP panel from Run 6 of the 1000 Bulls project presented in Naval-Sanchez et al 2020. This panel reduced standard errors and narrowed down confidence intervals for the ancient samples. Another consequence is that more single-source qpAdm models can now be rejected highlighting the abundance of hybridization. For our comparison to modern breeds, we still use the 700K dataset as it provides a set of different modern European cattle breeds.

I agree with the authors that qpAdm is likely to give quite a noisy estimate of ancestry here (likely explain part of the issue I mentioned above). Although qpAdm is good for model testing here for ancestry proportion the authors instead could use an explicit f4 ratio - this would allow them to specify a model which would make the result easier to interpret.

We have added ancestry estimates from f4 ratios to the manuscript and display them together with qpAdm and Struct-f4 (as suggested by reviewer #3) in our new Table 1. We decided to keep all three different estimates to illustrate that results are not consistent for all analyses. An additional feature of qpAdm is the possibility that two source models can be rejected and additional ancestries can be identified.

The interpretation of the different levels of allele sharing on X vs autosome being the result of sex-bias admixture is not very convincing. Could these differences simply be due to a low recombination rate on the X chromosome and/or lower effective population size, which would lead to less efficient purifying selection?

Following this comment (and another comment referring to the X chromosome analysis by reviewer #2), we decided to remove sex bias from the title of our study and add more information on the caveats of this analysis. While estimating ancestry on the X chromosome can be difficult, we also add that our patterns are consistent with what has been suggested based on ancient mitochondrial data (Verdugo et al 2019). For Neolithic Anatolia, it has been suggested that the insemination of domestic cows by auroch bulls has been intentional or even ritual (Peters et al 2012). A recent parallel archaeogenomic study also concluded sex-biased introgression from autosomal, X-chromosomal and Y-chromosomal data (Rossi et al 2024). As our results are consistent with these previous studies as well as the lower differentiation of modern breeds on the X chromosome (da Fonseca et al 2019), we still consider the general pattern of our results valid even if the exact extent of sex bias is difficult to assess.

The authors suggest that 2 pop model rejection in some domestic population might be due to indicine ancestry, this seems relatively straightforward to test.

We had already performed this analysis of modeling their ancestry from three sources using qpAdm. The results are shown in Supplementary Table S6 and we now refer to this more explicitly in the text: “The presence of indicine ancestry can be confirmed in a qpAdm analysis using three sources resulting in fitting models for all breeds (Supplementary Table S6).”

The first sentence of the paper is a bit long-winded, also dogs were domesticated before the emergence of farming societies.

We rephrased the first sentence to “Domestication of livestock and crops has been the dominant and most enduring innovation of the transition from a hunter-gathering lifestyle to farming societies.”

It would be good to be specific about the number of genomes and coverage info in the last paragraph of the intro.

This information is included in the first paragraph of the results section and we decided to not duplicate the numbers in the preceding introduction paragraph to retain a flow for the readers.

**Reviewer #2 (Public Review):**
Summary:In this paper, the authors investigated the admixture history of domestic cattle since they were introduced into Iberia, by studying genomic data from 24 ancient samples dated to ~2000-8000 years ago and comparing them to modern breeds. They aimed to (1) test for introgression from (local) wild aurochs into domestic cattle; (2) characterize the pattern of admixture (frequency, extent, sex bias, directionality) over time; (3) test for correlation between genetic ancestry and stable isotope levels (which are indicative of ecological niche); and (4) test for the hypothesized higher aurochs ancestry in a modern breed of fighting bulls.Strengths:Overall, this study collects valuable new data that are useful for testing interesting hypotheses, such as admixture between domestic and wild populations, and correlation between genome-wide aurochs ancestry and aggressiveness.

Thank you for highlighting the importance of our study and the potential of our dataset.

Weaknesses:Most conclusions are partially supported by the data presented. The presence of admixed individuals in prehistorical periods supports the hypothesized introgression, although this conclusion needs to be strengthened with an analysis of potential contamination. The frequency, sex-bias, and directionality of admixture remain highly uncertain due to limitations of the data or issues with the analysis. There is considerable overlap in stable isotope values between domestic and wild groups, indicating a shared ecological niche, but variation in classification criteria for domestic vs wild groups and in skeletal elements sampled for measurements significantly weakens this claim. Lastly, the authors presented convincing evidence for relatively constant aurochs ancestry across all modern breeds, including the Lidia breed which has been bred for aggressiveness for centuries. My specific concerns are outlined below.Contamination is a common concern for all ancient DNA studies. Contamination by modern samples is perhaps unlikely for this specific study of ancient cattle, but there is still the possibility of cross-sample contamination. The authors should estimate and report contamination estimates for each sample (based on coverage of autosomes and sex chromosomes, or heterozygosity of Y or MT DNA). Such contamination estimates are particularly important to support the presence of individuals with admixed ancestry, as a domestic sample contaminated with a wild sample (or vice versa) could appear as an admixed individual.

We thank the reviewer for this suggestion. Due to our low coverage data, we focused on estimating contamination from the mitochondrial data by implementing the approach used by Green et al (2008). We make the code for this step available on Github. While most samples displayed low levels of contamination, we identified one sample (moo013a) with a surprisingly high (~50%) level of contamination which was excluded from further analysis.

A major limitation of this study is uncertainty in the "population identity" for most sampled individuals (i.e., whether an individual belonged to the domesticated or wild herd when they were alive). Based on chronology, morphology, and genetic data, it is clear the Mesolithic samples from the Artusia and Mendandia sites are bona fide aurochs, but the identities of individuals from the other two sites are much less certain. Indeed, archeological and morphological evidence from El Portalon supports the presence of both domestic animals and wild aurochs, which is echoed by the inter-individual heterogeneity in genetic ancestry. Based on results shown in Fig 1C and Fig 2 it seems that individuals moo017, moo020, and possibly moo012a are likely wild aurochs that had been hunted and brought back to the site by humans. Although the presence of individuals (e.g., moo050, moo019) that can only be explained by two-source models strongly supports that interbreeding happened (if cross-contamination is ruled out), it is unclear whether these admixed individuals were raised in the domestic population or lived in the wild population and hunted.

The reviewer is pointing out an important topic, the unknown identity of the studied individuals. We have revised the text making clear that we do not know whether the individuals were hunted or herded. At the same time, their genomic ancestry speaks for itself showing that there was hybridization between wild and domestic and that different individuals carried different degrees of wild ancestry. In the revised version, we have added the unknown identity as well as the fact that our results can be affected by both, changes in human hunting and herding practices over time. Regardless of the exact identity of the individuals, our results can still be seen as (a) evidence for hybridization and (b) changes in human practices (hunting and/or herding) and their relationship to bovids over time.

Such uncertainty in "population identity" limits the authors' ability to make conclusions regarding the frequency, sex bias, and directionality of gene flow between domestic and wild populations. For instance, the wide range of ancestry estimates in Neolithic and Chalcolithic samples could be interpreted as evidence of (1) frequent recent gene flow or (2) mixed practices of herding and hunting and less frequent gene flow. Similarly, the statement about "bidirection introgression" (on pages 8 and 11) is not directly supported by data. As the genomic, morphological, and isotope data cannot confidently classify an individual as belonging to the domesticated or wild population, it seems impossible to conclude the direction of gene flow (if by "bidirection introgression" the authors mean something other than "bidirectional gene flow", they need to clearly explain this before reaching the conclusion.)

We have removed “bidirectional introgression” from the text and replaced it with the more neutral term “hybridization”. Furthermore, we used the revision to mention at several places in the text that it is not clear whether the sequenced individuals were hunted and herded and that the observed pattern likely reflects changes in both hunting and herding practices.

The f4 statistics shown in Fig 3B are insufficient to support the claim regarding sex-biased hybridization, as the f4 statistic values are not directly comparable between the X chromosome and autosomes. Because the effective population size is different for the X chromosome and autosomes (roughly 3:4 for populations with equal numbers of males and females), the expected amount of drift is different, hence the fraction of allele sharing (f4) is expected to be different. In fact, the observation that moo004 whose autosomal genome can be modeled as 100% domestic ancestry still shows a higher f4 value for the X chromosome than autosomes hints at this issue. A more robust metric to test for sex-biased admixture is the admixture proportion itself, which can be estimated by qpAdm or f4-ratio (see Patterson et al 2012). However, even with this method, criticism has been raised (e.g., Lazaridis and Reich 2017; Pfennig and Lachance, 2023). In general, detecting sex-bias admixture is a tough problem.

In response to this comment and another comment by reviewer #1, we decided to remove sex bias from the title. In the revised version of our study, we have now switched this analysis from f4 statistics to comparing f4 ratios between the X chromosome and autosomes (Figure 3). Furthermore, we have added more information on the caveats of this analysis citing the articles mentioned by the reviewer. At the same time, we highlight that our patterns are consistent with what has been suggested based on ancient mitochondrial data (Verdugo et al 2019). Unfortunately, the low coverage data does not allow to call Y chromosomal haplotypes which would also allow an analysis of the paternal lineage. But our results are consistent with additional examples from the literature: For Neolithic Anatolia, it has been suggested that the insemination of domestic cows by auroch bulls has been intentional or even ritual (Peters et al 2012) and there is a lower differentiation of modern breeds on the X chromosome (da Fonseca et al 2019). A recent parallel archaeogenomic study also concluded sex-biased introgression from autosomal, X-chromosomal and Y-chromosomal data (Rossi et al 2024). Similar to the broader hybridization signal, our interpretation does not depend on the estimates for single individuals as we describe the broader pattern. As our results are consistent with previous results based on other types of data, we still consider the general pattern of our results valid even if the exact extent of sex bias is difficult to assess.

In general, the stable isotope analysis seems to be very underpowered, due to the issues of variation in classification criteria and skeletal sampling location discussed by the authors in supplementary material. The authors claimed a significant difference in stable nitrogen isotope between (inconsistently defined) domestic cattle and wild aurochs, but no figures or statistics are presented to support this claim. Please describe the statistical method used and the corresponding p-values. The authors can consider including a figure to better show the stable isotope results.

In combination with updated tables, we have added a supplementary figure showing the stable isotope results (S9). In light of the reanalysis of the genetic data, we have reassessed the genetic models used to assign species in the stable isotope analysis. We have provided more details of the statistical methods used and the p-values are given in the supplementary materials. There is a significant difference in the nitrogen isotope values when comparing B. taurus and B. primigenius (identified on morphology) but no other comparisons are significant at the p = 0.05 threshold. The reviewer highlights what we have mentioned in the supplementary material regarding the varied skeletal elements used for stable isotope analysis and the difficulty of assigning a species identity (as this depends on what criteria are used; morphological or some kind of genetic threshold of ancestry). Indeed, how to identify the species is at the heart of the paper. Given that identity could be defined in many ways, we have used 3 different genetic models to reflect this and the morphological categories, to help explore different possible scenarios. The reviewer is correct to point out that some of this analysis is not helped by the variety of skeletal elements used, but we have been careful not to over-interpret the results. The only samples that have nitrogen values higher than one standard deviation from the mean are domestic cattle, so it is not unreasonable to suggest that only domestic cattle have high nitrogen isotope values.

**Reviewer #3 (Public Review):**
Summary:Günther and colleagues leverage ancient DNA data to track the genomic history of one of the most important farm animals (cattle) in Iberia, a region showing peculiarities both in terms of cultural practices as well as a climatic refugium during the LGM, the latter of which could have allowed the survival of endemic lineages. They document interesting trends of hybridisation with wild aurochs over the last 8-9 millennia, including a stabilisation of auroch ancestry ~4000 years ago, at ~20%, a time coincidental with the arrival of domestic horses from the Pontic steppe. Modern breeds such as the iconic Lidia used in bullfighting or bull running retain a comparable level of auroch ancestry.Strengths:The generation of ancient DNA data has been proven crucial to unravel the domestication history of traditional livestock, and this is challenging due to the environmental conditions of the Iberian peninsula, less favourable to DNA preservation. The authors leverage samples unearthed from key archaeological sites in Spain, including the karstic system of Atapuerca. Their results provide fresher insights into past management practices, and permit characterisation of significant shifts in hybridization with wild aurochs.

We thank the reviewer for their positive assessment of our work and for highlighting the strength and potential of the study.

Weaknesses:- Treatment of post-mortem damage: the base quality of nucleotide transitions was recalibrated down to a quality score of 2, but for 5bp from the read termini only. In some specimens (e.g. moo022), the damage seems to extend further. Why not use dedicated tools (e.g. mapDamage), or check the robustness by conditioning on nucleotide transversions?

We agree that using such a non-standard data preparation approach requires some testing. Since our main analyses are all based on f statistics, we compared f4 statistics and f4 ratios of our rescaled base quality data with data only using transversion sites. While estimates are highly correlated, the data set reduced to transversions produces larger confidence intervals in f4 ratios due to the lower number of sites. Consequently, we decided to use the rescaled data for all analyses displayed in main figures. We also prefer not to perform reference based rescaling as implemented in mapDamage as it might be sensitive to mapping bias (Günther & Nettelblad 2019).

- Their more solid analyses are based on qpAdm, but rely on two single-sample donor populations. As the authors openly discuss, it is unclear whether CPC98 is a good proxy for Iberian aurochs despite possibly forming a monophyletic clade (the number of analysed sites is simply too low to assess this monophyly; Supplementary Table S2). Additionally, it is also unclear whether Sub1 was a fully unadmixed domestic specimen, depleted of auroch ancestry. The authors seem to suggest themselves that sex-biased introgression may have already taken place in Anatolia ("suggesting that sex-biased processes already took place prior to the arrival of cattle to Iberia").

We expanded the discussion on this topic but removed the analysis of whether European aurochs form a clade due to the low number of sites. We do highlight that a recent parallel study on aurochs genomes confirmed that Western European aurochs form a clade, probably even originating from an Iberian glacial refugium (Rossi et al 2024). Even if minor structure in the gene pool of European aurochs might affect our quantitative results, it should not drive the qualitative pattern. The same should be the case for Sub1 as our tests would detect additional European aurochs ancestry that was not present in Sub1. The corresponding paragraph now reads:

“A limitation of this analysis is the availability of genomes that can be used as representatives of the source populations as we used German and British aurochs to represent western European aurochs ancestry and a single Anatolian Neolithic to represent the original domestic cattle that was introduced into Europe. Our Mesolithic Iberian aurochs contained too little endogenous DNA to be used as a proxy aurochs reference and all Neolithic and Chalcolithic samples estimated with predominantly aurochs ancestry (including the 2.7x genome of moo014) already carry low (but significant) levels of domestic ancestry. However, the fact that all of these aurochs samples carried P mitochondria strongly suggests that western European aurochs can be considered monophyletic. Furthermore, a recent parallel study also concluded that Western European aurochs all form a clade (27). The Anatolian Sub1 might also not be depleted of any European aurochs ancestry and could not fully represent the original European Neolithic gene pool as also indicated by qpAdm and Struct-f4 identifying small proportions of other Asian ancestries in some Iberian individuals.

While these caveats should affect our quantitative estimates of European aurochs ancestry, they should not drive the qualitative pattern as our tests would still detect any excess European aurochs ancestry that was not present in Neolithic Anatolia.”

Alternatively, I recommend using Struct-f4 as it can model the ancestry of all individuals together based on their f4 permutations, including outgroups and modern data, and without the need to define pure "right" and "left" populations such as CPC98 and Sub1. It should work with low-coverage data, and allows us to do f4-based MDS plots as well as to estimate ancestry proportions (including from ghost populations).

We thank the reviewer for this suggestion. We added Struct-f4 as an analysis but observed that it would not converge in an individual-based analysis due to the low coverage of most of our samples. We added Struct-f4 results for samples with >0.1X to the new Table 1, the results are similar to the results obtained using f4 ratios and (to a lower degree) the qpAdm results.

- In the admixture graph analyses (supplementary results), the authors use population groups based on a single sample. If these samples are pseudohaploidised (or if coverage is insufficient to estimate heterozygosity - and it is at least for moo004 and moo014), f3 values are biased, implying that the fitted graph may be wrong. The graph shown in Fig S7 is in fact hard to interpret. For example, the auroch Gyu2 from Anatolia but not the auroch CPC98 also from Anatolia received 62% of ancestry from North Africa? The Neolithic samples moo004 and moo014 also show the same shocking disparity. I would consider re-doing this analysis with more than a sample per population group

There seems to be some confusion relating to the sample identity in these figures. CPC98 is British and not Anatolian while Gyu2 is from the Caucasus and not Anatolia which would explain why they are different. Furthermore, moo004 is mostly of domestic ancestry while, moo014 is mostly of European aurochs ancestry according to our other analyses, which should explain why they also behave differently in this analysis. To avoid confusion and since this is a supplementary analysis from which we are not drawing any major conclusions, we decided to remove the graphs and the analysis from the study.

**Recommendations for the authors:**

**Reviewer #2 (Recommendations For The Authors):**
Fig 3A: The red regression line is misleading. It seems to show that the average aurochs ancestry fraction has been steadily decreasing since ~8000 years ago, but the "averaging" is not meaningful as not all samples necessarily represent domestic cattle remains and the sample size is rather small. In other words, the samples are just a small, random collection of domestic and wild animals, and the average ancestry is subject to large sampling noise. I would suggest removing the regression line (along with the associated confidence interval) in this figure. It would also be helpful to label the samples with their IDs and morphology in the plot for cross-reference with other figures. Also, it is said in the legend that "Modern Iberian breeds... are added around date 0 with some vertical jitter". Do the authors mean "horizontal jitter" instead?

Thank you for noticing this! We have removed the regression line and corrected the figure legend.

Fig 2 vs Fig 3A: are the error bars the same in these two plots? They seem to be highly similar, if not identical, but the legends read very differently ("95% confidence interval by block-jackknife vs. on standard error"). Please explain.

The figure legends have been corrected.

Fig 3B: What do the error bars in Fig 3B mean? 95% confidence interval or one standard error? Please clarify in the legend.

We have removed this figure and replaced it with a different way of displaying the results (now Figure 3). We ensured that the error bars are displayed consistently across figures.

According to the f4 statistics shown in Fig 1C and Fig 3B, moo012b carries a relatively high amount of domestic ancestry. How is this compatible with the observation in Fig 2 that this individual can be modeled with 100% aurochs (i.e., aurochs as the single source)? Does this simply reflect the low genome coverage?

moo012b is indeed one of the lowest coverage samples in our has at <0.02x sequencing depth. Even in our revised analysis using more sites, there is a discrepancy between the results of f4 statistics and qpAdm (suggesting mostly domestic ancestry) and f4 ratio suggesting mostly aurochs ancestry (Figure 1C and Table 1). We believe that this highlights the sensitivity of different methods to assumptions about the relationships of sources and potential “outgroups” which might not be well resolvable with low coverage data and in the presence of potentially complex admixture. Our general results, however, do not depend on the estimates for single individuals as our interpretations are based on the general pattern.

I don't fully understand the rationale behind the statement "However, at some point, the herding practices must have changed since modern Iberian breeds show approximately 20-25% aurochs ancestry". Can the stable ancestry fraction from 4000 years to the present (relative to the highly variable ancestry before) reflect of discontinuation of hunting rather than changes in herding practices?

We agree that this statement was not justified here, we rephrased the sentence to “In fact, from the Bronze Age onwards, most estimates overlap with the approximately 25% aurochs ancestry in modern Iberian cattle” and generally tried to make the text more nuanced on the issue of herding and hunting practices.

**Reviewer #3 (Recommendations For The Authors):**
Thanks for this interesting piece of work. The results are clearly presented, and I have no additional concerns other than those reflected in the public report, except perhaps:(i) trying to use more informative sample names (eg. including the date and location). It may facilitate reading without going back and forth to the table "Sample List".

We have now added a main table listing our post-Mesolithic samples together with their age, site and estimated aurochs ancestry proportions. We hope that his table makes it easier for readers to follow our sample IDs.

(ii) Briefly describe in the main the age of aurochs and Sub1 not generated in this study.

Fixed.